# Variability of polycyclic aromatic hydrocarbons and their oxidative derivatives in wintertime Beijing, China.

Atallah. El zein[1], Rachel E. Dunmore[1], Martyn W. Ward[1], Jacqueline F. Hamilton[1], Alastair C. Lewis[2]

[1]Wolfson Atmospheric Chemistry Laboratories, Department of Chemistry, University of York, York, YO10 5DD, United Kingdom
[2]National Centre for Atmospheric Science, University of York, York, YO10 5DD, United Kingdom

*Correspondence to*: Atallah. El zein (atallah.elzein@york.ac.uk); Alastair C. Lewis (ally.lewis@ncas.ac.uk)

**Abstract.** Ambient particulate matter (PM) can contain a mix of different toxic species derived from a wide variety of sources. This study quantifies the diurnal variation and nocturnal abundance of 16 Polycyclic Aromatic Hydrocarbons (PAHs), 10 Oxygenated PAHs (OPAHs) and 9 Nitrated PAHs (NPAHs) in ambient PM in central Beijing during winter. Target compounds were identified and quantified using Gas Chromatography – time of flight mass spectrometry (GC-Q-TOF-MS). The total concentration of PAHs varied between 18 and 297 ng m$^{-3}$ over 3 h daytime filter samples and from 23 to 165 ng m$^{-3}$ in 15 h night-time samples. The total concentrations of PAHs over 24 h varied between 37 and 180 ng m$^{-3}$ (mean: 97 ± 43 ng m$^{-3}$). The total daytime concentrations during high particulate loading conditions for PAHs, OPAHs and NPAHs were 224, 54, and 2.3 ng m$^{-3}$, respectively. The most abundant PAHs were fluoranthene (33 ng m$^{-3}$), chrysene (27 ng m$^{-3}$), pyrene (27 ng m$^{-3}$), benzo[a]pyrene (27 ng m$^{-3}$), benzo[b]fluoranthene (25 ng m$^{-3}$), benzo[a]anthracene (20 ng m$^{-3}$) and phenanthrene (18 ng m$^{-3}$). The most abundant OPAHs were 9,10-Anthraquinone (18 ng m$^{-3}$), 1,8 Naphthalic anhydride (14 ng m$^{-3}$) and 9-Fluorenone (12 ng m$^{-3}$) and the three most abundant NPAHs were 9-Nitroanthracene (0.84 ng m$^{-3}$), 3-Nitrofluoranthene (0.78 ng m$^{-3}$) and 3-Nitrodibenzofuran (0.45 ng m$^{-3}$). ∑PAHs and ∑OPAHs showed a strong positive correlation with the gas phase abundance of NO, CO, SO$_2$, and HONO indicating that PAHs and OPAHs can be associated with both local and regional emissions. Diagnostic ratios suggested emissions from traffic road and coal combustion were the predominant sources for PAHs in Beijing, and also revealed the main source of NPAHs to be secondary photochemical formation rather than primary emissions. PM$_{2.5}$ and NPAHs showed a strong correlation with gas phase HONO. 9-Nitroanthracene appeared to undergo a photo-degradation during the daytime and showed a strong positive correlation with ambient HONO (R=0.90, P<0.001). The lifetime excess lung cancer risk for those species that have available toxicological data (16 PAHs, 1 OPAH and 6 NPAHs) was calculated to be in the range 10$^{-5}$ to 10$^{-3}$ (risk per million people range from 26 to 2053).

## 1 Introduction

Outdoor air pollution contains a complex set of toxicological hazards and has become the largest detrimental environmental effect on human health (WHO/IARC., 2016). Exposure to outdoor high particulate loading of PM$_{2.5}$ (aerodynamic diameter less than 2.5 μm) is linked to harmful health effects, particularly affecting urban populations (Raaschou et al., 2013; Hamra et al., 2014). The major sources of PM$_{2.5}$ in urban areas are

incomplete combustion or gas-to-particle conversion, and they contain a varied mix of chemicals including inorganic ions, organic carbon and elemental carbon (Bond et al., 2004; Saikawa et al., 2009). Polycyclic Aromatic Hydrocarbons (PAHs) and their oxidative derivatives (Nitrated PAHs and Oxygenated PAHs) are one class of species with high toxic potency (Zhang et al., 2009; Jia et al., 2011; Wang et al., 2011a). PAHs released in the atmosphere come from both natural and anthropogenic sources; anthropogenic emissions include incomplete combustion of fossil fuels, agricultural burning, industrial and agricultural activities and are considered predominant (Ravindra et al., 2008; Zhang et al., 2009; Poulain et al., 2011; Kim et al, 2013; Abbas et al., 2018); natural contributions such as volcanic eruptions and forest fires are reported to be a less significant contributor to total emissions (Xu et al., 2006; Abbas et al, 2018).

Vapour phase PAHs can undergo gas phase reactions with oxidants in the atmosphere (including hydroxyl, ozone and nitrate radicals) leading to the generation of a range of nitrated-PAHs and oxygenated-PAHs (Atkinson et al., 1990; Atkinson and Arey., 1994; Sasaki, 1997). Atmospheric reaction with chlorine atoms in the presence of oxygen has also been suggested as a new formation pathway of OPAHs (Riva et al., 2015). OPAHs and NPAHs are often more toxic than the parent PAHs, showing a direct-acting mutagenicity on human cells (Durant et al., 1996; Hannigan et al., 1998; Purohit and Basu, 2000; Wang et al., 2011a; Benbrahim et al., 2012). Beside their formation in the gas phase, OPAHs and NPAHs can also be produced by heterogeneous reactions (Ringuet et al., 2012a, Jariyasopit et al., 2014; Zimmermann et al., 2013; Wenyuan et al., 2014; Keyte et al., 2013). Many of these derivatives can also be linked to primary emissions from motor vehicles and combustion processes (Rogge et al., 1993, Albinet et al., 2007a; Jakober et al., 2007; Shen et al., 2012; Nalin et al., 2016).

Many studies in different countries have focused on studying toxic organic pollutants in PM$_{2.5}$ because they fall within the respirable size range for humans (Sharma et al., 2007; Ringuet et al., 2012b, Farren et al., 2015). In the last decade, a major focus has been given to Chinese cities such as Shanghai, Beijing, Guangzhou, Tianjin, and Shenzhen because of their population growth and geographic peripheral expansion in manufacturing capacity and energy industries which are located throughout each of the city's manufacturing zones. This has made China the world leader in energy consumption, but also the world's highest emitter of PM$_{2.5}$ and PAHs (Lin et al., 2018; Zhang et al., 2009; Xu et al., 2006). The majority of previous studies have reported PAH concentrations in 24 h averaged samples during short-term and long-term measurements campaigns (Tomaz et al., 2016; Alves et al., 2017; Niu et al., 2017; Benjamin et al., 2014; Wang et al., 2011a). However, a long averaging period creates some limitations such as sampling artefacts, notably where changing atmospheric photolysis conditions (air humidity, temperature, wind direction, ozone or other oxidant concentrations) may have a significant influence on PAHs concentrations and oxidation rates (Albinet et al., 2007b; Albinet et al., 2009, Goriaux et al., 2006, Tsapakis and Stephanou., 2003; Tsapakis and Stephanou., 2007, Ringuet et al., 2012b). More intensive and higher frequency measurements in field campaigns have been suggested as a means to improve the positive matrix factorization model performance (Tian et al., 2017, Srivastava et al., 2018). A few studies have used twice daily (12 h) sampling (Albinet et al., 2008; Zhang et al., 2018; Farren et al., 2015; Ringuet et al., 2012b), obtaining limited information on variability in concentrations during the daytime and night-time (Tsapakis and Stephanou., 2007). Shorter time periods for sampling (3 h and 4 h) are still very limited (Reisen and Arey., 2004; Srivastava et al., 2018). Considering the above, this paper determines the

**Commented [A3]:** Response to short comment (SC1)

**Commented [A4]:** Response to short comment (SC1)

**Commented [A5]:** Response to short comment (SC1)

**Commented [A6]:** Referee 2: Specific comments: Line 58

**Commented [A7]:** Referee 2: Specific comments: Line 60

**Commented [A8]:** Red comments in this paragraph are the Response to short comment (SC1)

**Commented [A9]:** Response to short comment (SC1)

temporal diurnal and nocturnal variation of the PM$_{2.5}$-bound concentrations of PAHs, OPAHs and NPAHs from
the air of Beijing in China, it shows the role of photochemistry in the formation of OPAHs and NPAHs and
associate the fate and evolution of PAHs, OPAHs and NPAHs with the gas phase concentrations of other
pollutants (O$_3$, CO, NO, NO$_2$, SO$_2$, HONO), the cancer risk associated with inhalation of PM$_{2.5}$ was calculated.
This paper explores the feasibility of higher frequency sampling in Beijing, to support the identification of
emissions sources from diagnostic ratios and correlations with atmospheric gas pollutants. These measurements
also raise the potential importance of the chemical relationship between NPAHs and HONO which may impact
the HONO budget in the atmosphere and, if included, improve related models. This study comes after three

> **Commented [A10]:** Referee 1: Comment 1

years of declaring the anti-pollution action plan and strategy taken by the municipal government of Beijing and
published in September 2013 (Ministry of Ecology and Environment The People's Republic of China, Beijing
toughens pollution rules for cleaner air, 2013), trying to increase the number of days with good air quality index
by prohibiting coal combustion, promoting clean energy vehicles and public transport and helping industrial
transformation and upgrading to new technologies.

> **Commented [A11]:** Referee 2: Comment 5


## 2 Experimental Steps

### 2.1 Sampling site and method

The sampling setup shown in Fig. S1 was located at the Institute of Atmospheric Physics, Chinese Academy of

> **Commented [A12]:** Referee 2: specific comments: L.76

Sciences in Beijing (39°58'28" N, 116°22'15" E) as part of the Air Pollution and Human Health (APHH)
research programme. PM$_{2.5}$ filter samples were collected on the roof of a 2-storey building about 8m above
ground level using a High-Volume Air Sampler (Ecotech HiVol 3000, Victoria, Australia) operating at 1.33 m$^3$
min$^{-1}$. Daytime particles were collected every three-hours during high PM concentration levels, nine-hours at
low PM levels and over 15 h at night-time during 18 continuous days (22 November 2016 to 9 December 2016).
Fifty-seven samples in total were collected. The daytime sampling started at 8:30 in the morning and the filter
was changed every 3 h. During low particulate loading conditions, the daytime sampling started at 8:30 in the
morning for a sampling duration of 9 h. Night-time sampling began at ~17:30 and ended at 08:30 the following

> **Commented [A13]:** Referee 2: specific comments: L.82

day. Prior to sampling, the quartz filters (20.3 × 25.4 cm) (supplied by Whatman (Maidstone, U.K.)) were baked

> **Commented [A14]:** Referee 2: specific comments: L.83

at 550 °C for 5 h in order to eliminate any organic matter. After sampling, filters were wrapped in aluminium
foil, sealed in polyethylene bags and stored at -20 °C until extraction and analysis.

### 2.2 Extraction method and clean up

All collected samples were extracted using an Accelerated Solvent Extractor automated system (Dionex, ASE
350). Prior to extraction, 1/16$^{th}$ (surface area equivalent to 25.7 cm$^2$) of each filter was cut using a hole puncher
(Ø=27 mm) and for each batch of 6 samples, one sample was spiked with a mixture of two deuterated-PAHs
(Phenanthrene-D10; Pyrene-D10), two deuterated-OPAHs (9-Fluorenone-D8; 9,10-Anthraquinone-D8), and two
deuterated-NPAHs (1-Nitronaphthalene-D7; 3-Nitrofluoranthene-D9), as surrogate standards for PAHs, OPAHs
and NPAHs, respectively, with concentration on filters corresponding to 400 ng (40 µl, 10 ng µl$^{-1}$ in
Acetonitrile). All punched samples were cut to small pieces and packed into 5 mL stainless steel extraction cells.
Extractions were carried out in acetonitrile as follows: Oven at 120°C, pressure at 1500 psi, rinse volume 60%
and 60 s purge time for three consecutive 5 min cycles. Extracts (V=20ml) were evaporated to approximately 6
mL under a gentle stream of nitrogen before the clean-up step. All samples and blanks were purified on a SPE
silica normal phase cartridge (1g/6ml; Sigma Aldrich) to reduce the impacts of interfering compounds in the
matrix and to help maintain a clean GC injection inlet liner. After the clean-up step, the solution of each sample
was evaporated to 1 mL under a gentle stream of nitrogen at room temperature (20°C) and transferred to 1.5 mL
autosampler amber vial. Each concentrated sample was stored at 4°C until analysis. The average recovery
efficiencies calculated from surrogate standards ranged from 85% to 96% (Phenanthrene-d10: $95 \pm 9$ %; Pyrene-
d10: $101 \pm 7$ %; 9-Fluorenone-d8: $98 \pm 13$ %; 9,10-Anthraquinone-d8: $102 \pm 11$ %; 1-Nitronaphthalene-d7: 93
$\pm 8$ %; 3-Nitrofluoranthene-d9: $101 \pm 11$ %) and the target compounds concentrations were calculated
incorporating measured recovery efficiencies.

**2.3 Chemical standards**

The chemical compounds that have attracted the most attention in previous studies are the 16 priority PAHs and
their derivatives, defined by the United States Environment Protection Agency (EPA). The choice of the organic
compounds investigated in this study is based on those associated with the particle phase and commercially
available standards. All compounds are listed in Table 1 and were purchased from Sigma Aldrich, Alfa Aesar
and Santa Cruz Biotechnology in the UK and had a minimum purity of 98%. In parallel to individual standards,
a mixed solution of the 16 EPA PAHs (CRM47940, Supelco, Sigma Aldrich) of 10 µg ml$^{-1}$ in acetonitrile was
also used. Standard solutions for calibrations were prepared in acetonitrile (HPLC grade, 99.9% purity, Sigma
Aldrich). Deuterated compounds were supplied by C/D/N isotopes and distributed by QMX Laboratories Ltd
(Essex, UK).

**2.4 GC/MS Analysis**

Target compounds were quantified using a GC - accurate mass Quadrupole Time-of-Flight GC/MS system (GC
Agilent 7890B coupled to an Agilent 7200 Q-TOF-MS). Parent PAHs were separated in a 35 min analysis time
using a capillary HP-5MS Ultra Inert GC column (Agilent; 5%-Phenyl substituted methylpolysiloxane; length:
30 m, diameter: 0.25 mm, film thickness: 0.25 µm). Inlet injections of 1 µL were performed in pulsed splitless
mode at 320 °C using an automated liquid injection with the GERSTEL MultiPurpose Sampler (MPS). Helium
was used as a carrier gas at 1.4 mL min$^{-1}$. The GC oven temperature was programmed to 65 °C for 4 min as a
starting point and then increased to 185 °C at a heating rate of 40 °C min$^{-1}$ and held for 0.5 min, followed by a
heating rate of 10 °C min$^{-1}$ to 240 °C and then ramped at 5 °C min$^{-1}$ until 320 °C and held isothermally for
further 6 min to ensure all analytes eluted from the column. The MS was operated in Electron Ionisation (EI)
mode at 70 eV with an emission current of 35 µA. Calibration solutions were injected 3 times in the same
sequence for samples and covered the range from 1pg µL$^{-1}$ to 1000 pg µL$^{-1}$.
The method development for OPAHs and NPAHs was based on previous studies (Albinet et al., 2006; Albinet
et al., 2014; Bezabeh et al., 2003; Kawanaka et al., 2007) using Negative Chemical Ionisation (NCI) performed
at 155 eV and 48 µA, with methane (CH$_4$, research grade 5.5, Air Liquide) as reagent gas. Target compounds

**Commented [A15]:** Response to short comment (SC1)

**Commented [A16]:** Referee 2: specific comments: L.102

**Commented [A17]:** Response to short comment (SC1)

were eluted using the RXi-5ms (Restek GC column) with similar phase and characteristics to HP-5ms. Analysis
was performed in 29.2 min and the GC settings were as follows: 1 µL of each sample was injected in pulsed
splitless mode at 310 °C, Helium flow was set to 1.2 mL min$^{-1}$, the initial oven temperature of 70 °C was held
for 4 min, followed by a heating rate of 60 °C min$^{-1}$ until 190 °C and then raised to 270 °C at rate of 25 °C min$^{-1}$
and ended with 5 °C min$^{-1}$ until 320 °C, held for 10 min. A 10-point calibration curve within the range 0.5 pg
µL$^{-1}$ to 1000 pg µL$^{-1}$, was obtained with the correlation coefficients from the linear regression from 0.980 to

155 0.999.

## 2.5 Data analysis and error evaluation

Data acquisition were recorded and processed using the Agilent Qualitative and Quantitative analysis software.
Target compounds were isolated using Extracted-Ion Chromatograms (EIC) and identified by the combination
of retention time and mass spectral match against the calibration standards measured simultaneously within the
samples. The limit of detection (LOD) was defined as the valid lowest measurable peak response to peak noise
near the elution time of the target peak (S/N = 3) in a mix of standards solutions. As the chemical noise
increases during the analysis of real samples the Limit of Quantification (LOQ) was defined S/N=10. These
recommendations are in accordance with previous analytical studies (Nyiri et al., 2016; Ramírez et al., 2015).
LOD values were evaluated from standards solutions and ranged between 1 pg and 20 pg for PAHs, 0.01 pg and
0.2 pg (except 1-naphthaldehyde 0.5 pg) for OPAHs and 0.02 pg to 0.25 pg for NPAHs.
To determine any sources of contamination during sample preparation and the analytical procedure, the solvent
(acetonitrile) and field blanks (n=2) were analysed following the same procedure as for the samples (Extraction,
SPE, Evaporation). Most target compounds were found to be below LOD (S/N=3) or orders of magnitude (up to
$10^3$ - $10^4$) lower than was found in the samples. A small number of compounds found in field blanks (1,8-
Naphthalic anhydride, Benzo[a]fluorenone, 1-Nitronaphthalene, 9-Nitroanthracene) have a higher contribution
(4-30 %) to very few filters (2 to 5 samples) collected over a 3 h time period, if this was co-incident with low
particulate loading conditions. The contribution to each compound from field blanks has been corrected in the
final data.
We evaluated the precision of the method by calculating the relative standard deviations (%RSD) from replicate
analysis as shown in Table 1. For PAHs, the precision of sample replicates (n=10) during inter-day and intra-
day varied from 1.8% to 8.9% (mean 5.2%) and 1.2% to 8.7% (mean 3.4%), respectively. The %RSD average
for deuterium labelled compounds was about 3.6%. For OPAHs and NPAHs, two different concentrations of
standards were analysed (50 pg; n=6 and 400 pg; n=6); inter-day precision of 10 OPAHs gives an average
%RSD of 6.8% (range: 5.4 - 8.9%) and intra-day precision of 5.6% (3.2 - 7.8%). Similar to OPAHs,
repeatability and reproducibility between days for NPAHs varied from 3.9% to 8.4% (mean 5.5%) and 3.2% to
9.7% (mean 5.2%), respectively. Hence, the estimated random error quantified by the standard deviation of the
measurements did not exceed 7% on average. The systematic error may be due to the influence of the sample
matrix during the analysis sequence on the quantification step and the calibration offset. It was estimated to be a
maximum 10% from the measured recovery of the deuterium species (Garrido-Frenich et al., 2006). Therefore,
the overall estimated error, combining the precision and the systematic effects, is less than 20%.

**Commented [A18]:** Response to short comment (SC1) And Referee 2: Specific comments: L.88

**Commented [A19]:** Response to short comment (SC1) And Referee 2: Specific comments: L.157

Another source of error can be attributed to sampling artefacts and this has been discussed in previous studies
(Schauer, C. et al., 2003, Goriaux, M. et al., 2006, Tsapakis and Stephanou, 2003). The absence of an ozone
denuder to trap the gas phase oxidants may lead to an underestimation of the true values of PAHs due to
chemical decomposition. Therefore, data from long sampling times and under high ozone ambient
concentrations may be biased by sampling artefacts by more than 100 % (Schauer et al., 2003, Goriaux et al.,
2006). However, at low ozone levels, negative artefacts were considered not significant (Tsapakis and
Stephanou 2003), whilst, at medium ozone levels (30-50 ppb) PAHs values were underestimated by 30 %
(Schauer et al., 2003). In addition, heterogeneous reactions during particle sampling may occur only on the
monolayer surface coverage with limited diffusion of oxidants to the bulk particles (Keyte et al., 2013 and
references therein). Previous studies reported that the formation of NPAHs during high-volume sampling is not
significant and calculated to be < 3 % (Arey et al., 1988) and < 0.1 % (Dimashki et al., 2000).
Considering the role of ozone (always below 30 ppb in this study, with a mean value: $10.4 \pm 8.8$ ppb), in
addition to sampling time and temperature, the estimation of the negative sampling artefacts on our data range
between 10 and 20 %, with the highest error estimation attributable to longest sampling time (15h).

**Commented [A20]:** Referee 1: comment 3: L.83
And Response to short comment (SC1)

**3 Results and discussion**
**3.1 Temporal variations of PAHs, OPAHs and NPAHs in PM$_{2.5}$**
The volume of literature on PM$_{2.5}$ has rapidly increased over the last two decades and various disciplines have
contributed to improve understanding about source emissions, chemical composition, and impact on people's
behaviour and health. In China, the official air quality guidelines for PM$_{2.5}$ expressed as annual mean and 24 h
average are 35 µg m$^{-3}$ and 75 µg m$^{-3}$, respectively (WHO 2016; Ministry of Ecology and Environment The
People's Republic of China, 2012). During the sampling period of this study (Nov - Dec 2016), PM$_{2.5}$ was
measured every hour and ranged from 3.8 to 438 µg m$^{-3}$, with an average concentration of 103 µg m$^{-3}$. The
average 24 h PM$_{2.5}$ concentration was $108 \pm 82$ µg m$^{-3}$ (range: 10 - 283 µg m$^{-3}$), exceeding the 24 h limit value
on 10 of the 18 sampling days. Concurrent PM$_{2.5}$ concentrations were averaged to the filter sampling times (3 h,
9 h, 15 h) and are shown in Fig.6 and Fig. S3. The daily (24 h) concentration of Benzo[a]pyrene ranged from
4.46 to 29.8 ng m$^{-3}$ (average $15 \pm 8.9$ ng m$^{-3}$), exceeding the 24 h average limit value of 2.5 ng m$^{-3}$ for China
(Ministry of Ecology and Environment The People's Republic of China, 2012) on all of the 18 days of sampling
period.

**Commented [A21]:** Referee 2: Specific comments: L.164

**Commented [A22]:** Referee 2: Specific comments: L.166

**Commented [A23]:** Referee 2: Specific comments: L.169

**Commented [A24]:** Referee 2: Specific comments: L.168-169

**Commented [A25]:** Referee 2: Specific comments: L.214

Fig. 1 shows the measured concentrations of PAHs in the 3 h daytime samples ranging from 18 to 297 ng m$^{-3}$
(average $87.3 \pm 58$ ng m$^{-3}$) and from 23 to 165 ng m$^{-3}$ (average $107 \pm 51$ ng m$^{-3}$) in the 15 h night-time samples.
The 24 h total concentrations (combined results from daytime and night-time samples) of the 16 PAHs varied
between 37 and 180 ng m$^{-3}$ (average $97 \pm 43$ ng m$^{-3}$). PAHs derivatives showed the following trends: total
OPAHs concentrations varied from 3.3 to 55 ng.m$^{-3}$ (average: $26 \pm 16$ ng m$^{-3}$) in daytime and from 8.9 to 95 ng
m$^{-3}$ (average: $41.6 \pm 26$ ng m$^{-3}$) at night-time; OPAHs were approximately 25 and 14 times higher than average
NPAHs in the daytime (average: $1.03 \pm 0.74$ ng m$^{-3}$, range: 0.13-2.3) and night-time (average: $3.06 \pm 1.8$ ng m$^{-3}$,
range: 0.57-6.43), respectively.

**Commented [A26]:** SD in this paragraph has been added in response to Referee 2: comment 1 and Referee 2 Specific comments: L.171 and L.174

PHE (See Table 1 for abbreviations), FLT, PYR, BaA, CHR, BbF and BaP were the largest contributors to the total PAH concentration. 9-FLON, 9,10-ANQ and 1,8-NANY were the three major O-PAHs species. The most abundant NPAHs were 3-NDBF, 9-NANT and 3-NFLT. The temporal profile and contributions of each compound to the total concentration are shown in Fig. 2 and detailed in Table 2. The highest concentrations recorded in this study were in the day of 29 Nov 2016; concentrations of all target compounds in the particulate phase are displayed in Fig. S2. Some nitro-compounds (5-NAC, 1-NPYR, 6-NCHR, 6-NBaP) were below LOQ in a few samples while one oxy-compound (1,8-Naphthalic anhydride) was outside the dynamic range and limit of linearity of the calibration curve for samples with high mass loading (Table 2). Similar dominant compounds were found in different urban cities (Xi'an, Jinan, Beijing) of China (Bandowe et al. 2014, Zhang et al. 2018, Wang et al., 2011c). The average 24 h total PAH concentrations (97 ng m$^{-3}$) in this study was higher than the average value reported for Guangzhou city in the south of China (average 45.5 ng m$^{-3}$, from Liu et al., 2015), however, it was lower than average values reported for Xi'an city in winter (range 14-701 ng m$^{-3}$; average 206 ng m$^{-3}$ from Wang et al., 2006) and in the suburb of Beijing in winter (average 277 ng m$^{-3}$, from Feng et al., 2005). Our average value (97 ng m$^{-3}$) was comparable to the reported values in a recent study (Feng et al., 2019) at the campus of Peking University health science centre, a short distance from our sampling site (~1 mile), where the authors reported a total PAHs average concentration in winter Beijing (2014 - 2015) of 88.6 ± 75 ng m$^{-3}$. The lower average concentration of total PAHs reported in this study and Feng et al., (2019) can potentially be attributed to the efforts from municipal government to improve air quality and control emissions by reducing combustion sources in the intervening years. The urban location in this study (Fig. S1) was surrounded by busy roads, residential buildings, an underground railway, restaurants and further afield thermal power stations. PAHs concentrations are anticipated to decline closer to the mountains in the North and West of Beijing due to air mass trajectory, aging and distance from emission sources. Results from this study can be considered representative (within the margin of error) of the urban area in Beijing including districts such as Chaoyang, Haidian, Fengtai, Xicheng, Dongcheng, Shijingshan covering an approximate population of 12 million. Future studies in less populated districts and different areas of the metropolitan of Beijing would be helpful for comparison of population exposures.

Concentrations of PAHs in PM$_{10}$ (range: 3.2 - 222.7 ng m$^{-3}$) in Beijing were found in previous studies to be lower than in PM$_{2.5}$ (Wang et al., 2011c). The concentration of PAHs in this study were much lower than reported in certain other megacities, for example, Delhi, India in winter season 2003 (range: 948-1345 ng m$^{-3}$; mean: 1157 ± 113 ng m$^{-3}$ from Sharma et al., 2007) and Mexico City, Mexico in October 2002 (range: 60-910 ng m$^{-3}$; mean: 310 ng m$^{-3}$ from Marr et al., 2004). Average concentrations for total PAH in the first 3 h filter of the day (8:30-11:30 am; Monday to Friday; mean: 112 ng m$^{-3}$) were 1.5 times higher than the rest of the day, and 1.6 times higher than the same first 3 h on a Sunday. A potential reason of the elevated concentrations in the morning hours is due to the rush hour traffic during working days, coupled to a period of shallow boundary layer.

The mean total concentrations in Table 2 for the 3 h integration samples of OPAHs and NPAHs were 28.7 ± 21 ng m$^{-3}$ (range: 1.8 - 87.9 ng m$^{-3}$) and 1.17 ± 1 ng m$^{-3}$ (range: 0.15 - 3.92 ng m$^{-3}$). Average night-time was 41.6 ng m$^{-3}$ (OPAHs) and 3.06 ng m$^{-3}$ (NPAHs), concentrations which were 2.6 and 35 times lower than the average total PAH in the night samples. The ratios of mean concentrations of PAH divided by concentration of OPAH

**Commented [A27]:** Response to Referee 1: comment 2: Line 76-86 and Referee 2 Specific comments: L.193

**Commented [A28]:** Response to Referee 1: comment 5: Line 369 and Referee 2 Comment 6

**Commented [A29]:** SD, Response to Referee 2 comment 1

and NPAH for the 3 h samples were 3 and 74. Ratios of combined daytime and night-time samples (24 h) were on average 2.9 (range 1.9 - 4.6) for ∑PAHs/∑OPAHs and 47.4 (range 25 - 79) for ∑PAHs/∑NPAHs. Lower ratios were reported from a winter study in Xi'an – China, where ∑PAHs/∑OPAHs ranged from 1.75 to 1.86 and ∑PAHs/∑NPAHs ranged from 34 to 55.2. On the other hand, similar trends to our study were recorded for ∑PAHs/∑OPAHs in Europe such as Athens in Greece in winter (ratio 28.9/6.9 = 4.2) (Andreou and Rapsomanikis., 2009) and Augsburg in Germany in winter (ratio 11/3.2 = 3.4) (Pietrogrande et al., 2011). Further monitoring studies are needed to confirm trends of NPAHs in China.

### 3.2 Diagnostic ratios to identify emission sources

The concentration ratios between different PAHs are widely used to assess and identify pollution emission sources (Tobiszewski and Namieśnik., 2012 and references therein). The ratios of FLT/(FLT + PYR) and IcdP/(IcdP + BghiP) isomer pairs are commonly used to distinguish between emission sources such as coal/biomass burning or the incomplete combustion of petroleum. Values of FLT/(FLT + PYR) and IcdP/(IcdP + BghiP) higher than 0.5 indicate dominance of a coal/biomass burning source. Values of FLT/(FLT + PYR) between 0.4 and 0.5 and IcdP/(IcdP + BghiP) between 0.2 and 0.5 suggest a higher influence from fossil fuel combustion. Values of FLT/(FLT + PYR) less than 0.4 and IcdP/(IcdP + BghiP) less than 0.2, are mostly related to incomplete combustion (petrogenic origin) (Yunker et al., 2002; Pio et al., 2001). The measured ratios in this study are shown in Fig. 3 and ranged from 0.53 to 0.67 (mean 0.56) during the day (3 h and 9 h samples), while at night (15 h samples) varied between 0.51 and 0.54 (mean 0.52) indicating primary emissions from coal and biomass burning. Lower values were observed for IcdP/(IcdP + BghiP); daytime ratios were between 0.39 and 0.5 (3 h and 9 h samples) indicated the dominance of petroleum combustion. At night, the ratio in most samples was slightly higher than 0.5, with some values below, suggesting mixed sources with likely higher contributions coming from residential heating using coal and wood at night.

As shown in Fig. 4, other ratios can be useful to confirm the contribution from local traffic and to discriminate vehicle emissions such as BaP/BghiP, FLU/FLU+PYR and BaP/BaP+CHR (Tobiszewski and Namieśnik., 2012 and references therein). The BaP/BghiP ratios were significantly higher than 0.6 indicating a major influence from road traffic, while FLU/FLU+PYR ratios suggested a predominant petrol contribution (ratio < 0.5) instead of diesel engines (ratio > 0.5). Results shown in Fig. 4 identify traffic emissions and in particular petrol engines as the major emitter of PAHs. In $PM_{2.5}$, the 5- and 6-rings PAHs species (BaP, IcdP, BghiP) were previously attributed to petrol engines, while lower molecular weight with 3-rings (ACY, AC, FLU, PHE, ANT) and 4-rings (FLT, PYR, BaA, CHR) were closely related to diesel vehicle emissions (Chiang et al., 2012; Wu et al., 2014 and references therein). Previous studies in Beijing and Guangzhou in China suggested similar contributions from coal and petroleum combustion, focusing on vehicular traffic (petrol and diesel) as potential sources for PAHs (Gao and Ji., 2018; Liu et al., 2015; Wu et al., 2014, Niu et al., 2017).

NPAHs can be used to track the photochemistry of PAHs with OH and $NO_3$ radicals, both of which can generate secondary photochemical products of NPAHs and OPAHs from primary PAH emissions (Zhang et al., 2018; Ringuet et al., 2012b; Wang et al., 2011a). 1-NPYR originates mainly from primary emissions and in particular from diesel vehicles (Keyte et al., 2016, Schulte et al., 2015), whilst 2-NFLT has been reported to be absent in

direct combustion emissions, instead produced from the gas-phase reactions of FLT with OH radicals in presence of NOx during the day or $NO_3$ radicals at night. 2-NPYR comes solely from the reaction of PYR with OH radicals (Ramdahl et al., 1986; Arey et al., 1986; Atkinson et al., 1987; Ciccioli et al., 1996). Accordingly, the ratio 2-NFLT/1-NPYR has been widely used as diagnostic, with a value greater than 5 indicating a major contribution from photochemical processes, whilst a ratio value less than 5 means an important contribution from direct emissions (Albinet et al., 2008; Wang et al., 2011a; Ringuet et al., 2012b; Bandowe et al., 2014; Tomaz et al., 2017; Zhang et al., 2018).

In this study, the 2-NFLT was not quantified because the standard compound was not commercially available, subsequently, we have used 3-NFLT isomer as a substitution of 2-NFLT. PAH isomer pairs (Table 1) in standard mixtures showed similar sensitivities for each concentration used, therefore, we assume an equal sensitivity for 2-NFLT and 3-NFLT during analysis. A previous study reported that the concentration of 3-NFLT compared to 2-NFLT is relatively very low in ambient air (Bamford et al., 2003); in addition, the separation of both isomers (2- and 3-NFLT) using the most common GC-MS column for PAHs separation, HP-5ms and DB-5ms, was not possible (Zhang et al., 2018; Bandowe et al., 2014; Ringuet et al., 2012b; Albinet et al., 2008). Hence, we assume that the sum of 2- and 3-NFLT is closely representative of the original ratio 2-NFLT/1-NPYR. Therefore, we adopted the ratio 2+3NFLT/1-NPYR, which varied between 4 and 19 during the daytime (mean: 12) and from 3.6 to 30.4 in the night-time (mean: 8.8) (Fig. S3). Most daytime values exceeded significantly the benchmark ratio of 5, while at night-time the average value was lower. These results indicate the predominance of OH-radicals-initiated reactions controlling the formation of 2-NFLT in presence of $NO_2$ and sunlight.

**3.3 Correlation with gaseous pollutants**

$O_3$, CO, NO, $NO_2$, $SO_2$ and HONO were also measured at the same site location as the $PM_{2.5}$ sampling. Inlets were installed outside lab containers at approximately 3-4 m above ground (Fig. S1). Online measurements of the gas phase species have been time-averaged to the filter sampling times. No correlations of significance were seen between PAHs and meteorological parameters (Relative Humidity and Temperature) as shown in Table S1.

∑PAHs and ∑OPAHs had a similar strong positive correlation (R= 0.82 to 0.98) in the 9 h and 15 h samples with CO, NO, $NO_2$, $SO_2$ and HONO (Table S1). NO is known as an effective tracer for local traffic emissions, it and behaves as a short-lived intermediate (Bange 2008, Janhäll et al., 2004). CO is mainly produced from incomplete combustion and has a relatively long atmospheric lifetime (3 months on average) and undergoes long-range transport (Peng et al., 2007 and references therein). The high correlations with primary pollutants such as NO and CO during the daytime and night-time indicate that PAHs and OPAHs are primarily emitted from local sources may also be associated with regional scale emissions. Significant correlations were observed with $SO_2$, a pollutant mostly emitted from power plants outside the city (Lee et al., 2011). This strong relationship with $SO_2$ could be explained by the contribution of anthropogenic sources such as the Beijing Taiyanggong thermal power station (39°58′42″N 116°26′19″E), and is consistent with the domain air masses arriving at the sites from the North East for much of the time (Fig. S4).

In contrast, most of the 3 h day samples showed only moderate correlations (R=0.38 to 0.74) except for HONO where significant correlations (R=0.87 to 0.94) were observed with ∑PAHs, ∑OPAHs and ∑NPAHs (Fig. 5;

Table S1). Furthermore, HONO was significantly correlated with $PM_{2.5}$ during the daytime (Fig. 5) and some significant chemical link between HONO emissions and ambient particles ($PM_{2.5}$) is implied. A similar conclusion was drawn from recent study in Beijing (Zhang et al., 2019) which suggested a potential chemical relationship between HONO and haze particles ($PM_{2.5}$) and proposed a high contribution from vehicle emissions to the night-time HONO.

For NPAHs, as shown in Table S1, no significant correlation was found in 3 h and 15 h time sampling resolution, except with HONO, where a significant difference between day and night were observed. Surprisingly, the 9 h time resolution showed a strong correlation with CO, NO, $NO_2$ and $SO_2$, potentially suggesting a direct emission of NPAHs. More likely these correlations arise because of a formation delay of NPAHs that is smoothed out by the longer daytime sampling period. In a previous study, Zimmermann et al., (2013) reported the formation of NPAHs from the heterogeneous interaction of ambient particle bound-PAHs with atmospheric oxidant. In line with the observed high values for the ratio 2+3NFLT/1-NPYR (section 3.2) and the trace levels of NPAHs concentrations in the atmosphere; the secondary formation of NPAHs by gas phase reactions followed by adsorption on particles and in parallel the heterogeneous formation on the surface of particles is supported rather than primary emissions.

HONO plays a key role in tropospheric photochemistry, however its sources and their relative contributions to ambient HONO are still unclear, especially in the daytime. To help understand the mechanism of HONO formation in the atmosphere, each NPAHs compound has been correlated with HONO concentrations. The available data in Table S2 shows diurnal and nocturnal differences for individual correlation of NPAHs with HONO with the exception for 1-NPYR, which originates mainly from primary emissions and shows a strong correlation during the day and night. 9-Nitroanthracene had distinctive behaviour, accumulating during the night and appearing to undergo a photo-degradation during the daytime (Fig. 6). As shown in Table S2, 9-Nitroanthracene showed a strong positive correlation with HONO (R=0.90, P<0.001) in the daytime while no significant relationship was found at night-time (R=0.15, P>0.05). This suggests 9-nitroanthracene as a possible source of HONO during the daytime via the OH radical-initiated reaction leading to OH (ortho) addition and followed by intramolecular hydrogen transfer from the phenolic hydroxyl group to the nitro group.

There was a significant positive correlation between ANT and 9-NANT (R= 0.90, 3 h; R=0.94, 9 h; R=0.90, 15 h; P≤0.001), which may be an indication that 9-NANT is closely related to ANT. In this respect, additional simulation chamber measurements of the gas phase reaction of ANT with $NO_3$ radicals and for 9-Nitroanthracene with OH radicals in presence of light and under different atmospheric parameters are required for more precise assessment.

## 3.4 Exposure assessment

The toxicity equivalency factor (TEF) represents an estimate of the relative toxicity of a chemical compared to a reference chemical. For PAHs, Benzo[a]pyrene was chosen as the reference chemical because it is known as the most carcinogenic PAH (OEHHA., 1994, 2002) and is commonly used (Albinet et al., 2008; Tomaz et al., 2016; Alves et al., 2017; Bandowe et al., 2014; Ramírez et al., 2011) as an indicator of carcinogenicity of total PAHs.

**Commented [A36]:** Response to Referee 2: Comment 4

**Commented [A37]:** Response to Referee 2: Comment 4

**Commented [A38]:** Response to Short Comment (SC1)

The toxicity of the total PAHs expressed as BaP equivalents ($BaP_{eq}$) is calculated from the TEFs of each target compound (Table S3) multiplied by its corresponding concentration Eq. (1):

$$\sum [BaP]_{eq} = \sum_{i}^{n=1} (C_i \; x \; TEF_i) \qquad (1)$$

where $C_i$ correspond to the concentration of individual target compound (PAHs, OPAHs and NPAHs) in ng m$^{-3}$.

A widely applied procedure from the Office of Environmental Health Hazards Assessment (OEHHA) of the California Environmental Protection Agency (CalEPA) and also the World Health Organisation (WHO) was used here to evaluate and calculate the potential of contracting cancer from inhalation and exposure to PM$_{2.5}$-bound PAHs; commonly known as the lifetime excess cancer risk (ECR) Eq. (2).

$$ECR = \sum [BaP]_{eq} \; \times \; UR_{BaP} \qquad (2)$$

where two values are mostly used for $UR_{[BaP]}$ (1.1x10$^{-6}$ (ng m$^{-3}$)$^{-1}$ (OEHHA., 2002, 2005) and 8.7x10$^{-5}$ (ng m$^{-3}$)$^{-1}$ (WHO., 2000)); Eq. (2) describes the inhalation unit risk associated with high probability of contracting cancer when exposed continuously to 1 ng m$^{-3}$ of $BaP_{eq}$ concentration over a lifetime of 70 years.

As shown in Table 3, the $BaP_{eq}$ concentration includes the sum of 16 PAHs, 1 OPAH and 6 NPAHs, with the cancer risk evaluated using different sampling times according to CalEPA and WHO guidelines. The risk values may be underestimated due to lack of toxicity data for OPAHs and because our assessment excludes the gas phase contributions i.e. are only based on the health risk evaluation of the particulate phase. The average 24 h $BaP_{eq}$ for the whole sampling period was 23.6 ± 12.4 ng m$^{-3}$ (Table 3). As shown in Table 2, 6-NCHR has not been quantified in all samples, its contribution to the total $BaP_{eq}$ is relatively high (mean: 8%, range: 1 - 24%) in comparison with the three major contributor from the PAH group: BaP (mean: 47.5%, range: 24 - 64%), DahA (mean: 17.8%, range: 10 - 32%) and BbF (mean: 10.1%, range: 7 - 21%). In this study, the ECR attributable to all polycyclic aromatic compounds (PACs) in urban air of Beijing ranged from 10$^{-5}$ to 10$^{-3}$ > 10$^{-6}$ (Table 3) suggesting an elevated potential cancer risk for adults (Chen and Liao., 2006; Bai et al., 2009).

It is worth noting that inhalation exposure is not the only risk related to PAHs and cancer in humans. Other sources of exposure include dermal contact and ingestion of the re-suspended dusts in matrices such as road dusts and soils all of which increase the risk value for urban residents (Wang, et al., 2011b; Wei et al., 2015). In this study, the 24 h average estimated cancer risk (Table 3) from inhalation exposure to ambient PM$_{2.5}$ based on CalEPA and WHO guidelines were 2.6 x 10$^{-5}$ and 2.05 x 10$^{-3}$, respectively. Using the highest calculated ECR (2.05 x 10$^{-3}$) gives an estimate of 2027 additional cancer cases per million people exposed (29 cases/year) in comparison to the estimate based on CalEPA of 26 persons (0.37 cases/year).

ECR trends were reported in previous studies from Beijing and other populated area (Bandowe et al., 2014; Alves et al., 2017; Ramírez et al., 2011; Jia et al., 2011, Liu et al., 2015, Song et al., 2018, Feng et al., 2019). In this study we considered the combination of all samples (n=54) to estimate the average 24 h cancer risk

Commented [A39]: Response to referee 2 Comment 1

Commented [A40]: Response to Referee 2: Comment 4

Commented [A41]: Response to Referee 2: Comment 4

Commented [A42]: Response to Referee 2: Comment 4

$(\sum[BaP]_{eq}=23.6 \pm 12$ ng m$^{-3}$; range 8 – 44 ng m$^{-3}$) and compare it with previous studies. An average value of 17 ng m$^{-3}$ (range 2-64 ng m$^{-3}$) was reported for Xi'an for the whole year between July 2008 and August 2009 (Bandowe et al., 2014). After considering the same winter period (November and December) as in our study, the average values reported for Xi'an city (31-33 ng m$^{-3}$) were higher than our results. In contrast, our average value was comparable to those reported in a recent study in Beijing, ranging from 21 to 38 ng m$^{-3}$ in cold months (Feng et al., 2019), whilst in the previous study of Chen et al. 2017, they reported an average of 31.4 ng m$^{-3}$ for outdoor air in Beijing in winter. Lower and more varied values have been also reported in Beijing in winter. Liu et al. (2007) reported an average BaP$_{eq}$ concentration of 13.0 ng m$^{-3}$ and 27.3 ng m$^{-3}$ at two sampling sites on Peking University campus and 82.1 ng m$^{-3}$ for samples collected from busy road street. It is clear that direct comparison with Beijing air from other studies is limited due to the variable number of compounds considered in each study and the differences in sampling sites and sampling periods. Other areas of uncertainty include TEF reference values and the range of BaP UR which were extrapolated from animal bioassays with limited evidence regarding the carcinogenicity to humans.

Seasonal variability is also crucial in estimating BaP$_{eq}$ concentrations; it has been shown that BaP$_{eq}$ values in cold months are always higher than warm months due to the increase in coal combustion, central and residential heating, lower photochemical transformation and lower volatilisation of gases favouring particle formation in winter. Previous observations in Beijing recorded $\sum[BaP_{eq}]$ of 11.1 ng m$^{-3}$ in autumn (Jia et al., 2011) and 11.0 ng m$^{-3}$ in warm months (April to June) (Feng et al., 2019). In comparison with Guangzhou city (south of China), BaP$_{eq}$ was 9.24 ng m$^{-3}$ in winter and reported to be 1.6 and 6.2 times greater than autumn and summer, respectively (Liu et al., 2015). Our results were considerably higher than those estimated for western European cities during the winter, such as Grenoble: 1.4 ng m$^{-3}$ (Tomaz et al., 2016), Oporto: 3.56 ng m$^{-3}$, Florence: 1.39 ng m$^{-3}$ and Athens: 0.43 ng m$^{-3}$ (Alves et al., 2017). ECR values estimated for each city were 31 (Grenoble), 6.6 (Oporto), 17 (Florence) and 54 (Athens) times lower than our ECR estimation. Lower ECR levels in western European cities were attributed to cleaner energy sources, less densely populated cities, waste exporting and recycling and more effective environmental regulations.

**Commented [A43]:** Response to Short Comment (SC1)

## 4 Conclusions

Temporal variations and chemical composition of PM$_{2.5}$ were measured in Beijing-China from 22 November 2016 to 9 December 2016, focusing in particular on the diurnal and nocturnal chemical formation of PAHs, OPAHs and NPAHs. The 24 h average concentration of PM$_{2.5}$ was 108 µg m$^{-3}$ ranging from 10 to 283 µg m$^{-3}$, exceeding the 24 h limit value for China on 10 out of 18 sampling days. The 24 h concentrations of $\sum$PAH$_{16}$ varied between 37 and 180 ng m$^{-3}$ (average 97 $\pm$ 43 ng m$^{-3}$), while $\sum$OPAH$_{10}$ ranged from 13 to 70 ng m$^{-3}$ (average 35.6 $\pm$ 19 ng m$^{-3}$) and $\sum$NPAH$_9$ from 0.87 to 4.4 ng m$^{-3}$ (average 2.29 $\pm$ 1.2 ng m$^{-3}$). Daytime concentrations during pollution episodes for PAHs, OPAHs and NPAHs were 224, 54, and 2.3 ng m$^{-3}$, respectively. The daily concentration of Benzo[a]pyrene exceeded the 24 h average limit value of 2.5 ng m$^{-3}$ for China on all sampling days in this study, indicating elevated risk of disease among inhabitants.

**Commented [A44]:** SD Response to Referee 2 comment 1

**Commented [A45]:** Response to Referee 2: Comment 4

Diagnostic ratios of different species were used to distinguish between possible emission sources of PAHs. Coal combustion and road traffic emissions (petrol engines) were found overall to be the two dominant sources. In addition, high ratios of 2+3Nitrofluoranthene/1-Nitropyrene indicated significant secondary formation of NPAHs, especially in daytime via the OH radical-initiated reaction pathway.

Commented [A46]: Response to Referee 2: Comment 4

PAHs and OPAHs concentrations were correlated with CO, NO, $NO_2$, $SO_2$ and HONO, indicating that both are associated with local and regional primary emissions and in particular to traffic sources. Correlations seen previously between $PM_{2.5}$ and HONO suggested a possible links and a potential source of HONO that would affect the budget of HONO and OH radicals. The strong positive correlation between individual NPAHs and HONO during daytime was also suggestive of a potential link between these two classes of chemicals in air. One of the dominant NPAHs, the 9-Nitroanthracene had distinctive behaviour, accumulating at night and photodegrading in daytime.

Commented [A47]: Response to Referee 2: Comment 4

The lifetime excess cancer risk attributable to the summation of polycyclic aromatic compounds measured here and associated with $PM_{2.5}$ inhalations in Beijing was in the range of $10^{-3}$ according to WHO guidelines, confirming that there is statistically elevated risk of contracting cancer from this class of pollutants in this location.

*Author contributions:* AE led the analysis and prepared the manuscript with contributions from all authors. ACL and JFH contributed to the analysis, interpretation and writing of the paper. RED provided the data on the gas phase measurements and collected the filter samples in the field. MWW supported laboratory chemical analysis on the GC-Q/ToF-MS. All authors contributed to the corrections of the paper.

*Competing interests.* The authors declare that they have no competing interests.

*Acknowledgements*: Authors gratefully acknowledge the U.K. Natural Environment Research Council for funding Air Pollution and Human Health programme, reference: NE/N007115/1 and NE/N006917/1. We thank Leigh Crilley and Louisa Kramer from the University of Birmingham for provision of HONO data, funded through the APHH AIRPRO and AIRPOLL projects references NE/N007115/1 and NE/N006917/1. Authors gratefully acknowledge the vital contributions of Prof. Pingqing Fu and his staff at the Institute of Atmospheric Physics, CAS in Beijing for enabling the field observations and providing resources to support the wider project.

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

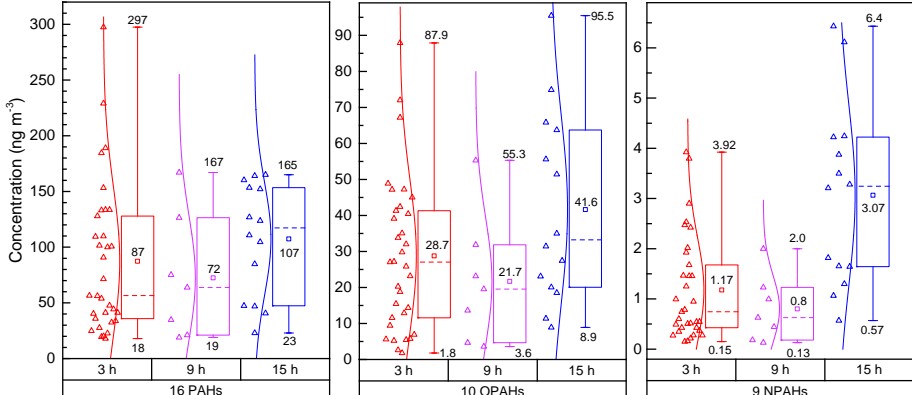

**Figure 1.** Concentrations of ∑PAHs, ∑OPAHs and ∑NPAHs in PM$_{2.5}$ samples during the daytime (3 h; 9 h) and night-time (15 h). Box plot represents the 25th and 75th percentiles range of the observed concentrations and the bottom and top lines indicate minimum and maximum concentrations. Square symbols represent the mean concentration, and the short dash line within the boxes represent the median. Empty Triangles correspond to the data measured over 3 h, 9 h and 15 h samples. The lines between data points and boxes reflect a normal distribution curve.

Table 1. List of measured PAHs, OPAHs and NPAHs and their Abbreviations. Compounds are listed in order of elution.

| Compound/Formula | Abbreviation | Accurate Mass (*m/z*) | %RSD | |
|---|---|---|---|---|
| 16 PAHs | | Monitored ions in EI mode | Inter-day | Intra-day |
| Naphthalene/C$_{10}$H$_8$ | NAP | 128.0628-127.0543-102.0464 | 4.6 | 3.2 |
| Acenaphthylene/C$_{12}$H$_8$ | ACY | 152.0629-151.0546-126.0463 | 4.1 | 2.1 |
| Acenaphthene/ C$_{12}$H$_{10}$ | AC | 153.0705-154.0779-152.0634 | 5.5 | 6.1 |

Commented [A51]: Response to Referee 2: Specific comments: L.662

| Compound/Formula | Abbrev. | Monitored ions | | |
|---|---|---|---|---|
| Fluorene/ $C_{13}H_{10}$ | FLU | 166.0782-165.0708-164.0621 | 4.0 | 2.9 |
| Phenanthrene/$C_{14}H_{10}$ | PHE | 178.0789-176.0626-152.0622 | 4.6 | 3.0 |
| Anthracene/$C_{14}H_{10}$ | ANT | 178.0787-176.0627-152.0620 | 4.7 | 4.2 |
| Fluoranthene/$C_{16}H_{10}$ | FLT | 202.0788-200.0626-101.0388 | 1.8 | 4.5 |
| Pyrene/$C_{16}H_{10}$ | PYR | 202.0788-200.0626-101.0389 | 3.2 | 1.9 |
| Benzo[a]anthracene/$C_{18}H_{12}$ | BaA | 228.0927-226.0783-101.0388 | 6.2 | 1.2 |
| Chrysene/$C_{18}H_{12}$ | CHR | 228.0943-226.0784-101.0387 | 6.0 | 2.6 |
| Benzo[b]fluoranthene/$C_{20}H_{12}$ | BbF | 252.0941-250.0784-126.0467 | 4.7 | 2.0 |
| Benzo[k]fluoranthene/$C_{20}H_{12}$ | BkF | 252.0940-250.0783-126.0468 | 8.9 | 8.7 |
| Benzo[a]pyrene/$C_{20}H_{12}$ | BaP | 252.0940-250.0783-126.0466 | 5.2 | 2.3 |
| Indeno[1,2,3-cd]pyrene/$C_{22}H_{12}$ | IcdP | 276.0939-274.0783-138.0467 | 7.2 | 2.6 |
| Dibenz[a,h]anthracene/$C_{22}H_{14}$ | DahA | 278.1097-276.0941-139.0545 | 7.7 | 4.3 |
| Benzo[ghi]perylene/$C_{22}H_{12}$ | BghiP | 276.0942-274.0783-138.0467 | 5.4 | 2.6 |
| 10 OPAHs | | Monitored ions in NCI mode | | |
| 1,4-Naphtoquinone/$C_{10}H_6O_2$ | 1,4-NAQ | 158.0420 | 6.3 | 5.1 |
| 1-Naphthaldehyde/$C_{11}H_8O$ | 1-NALD | 156.0557 | 8.9 | 7.8 |
| 9-Fluorenone/$C_{13}H_8O$ | 9-FLON | 180.0639 | 5.7 | 6.2 |
| 9,10-Anthraquinone/$C_{14}H_8O_2$ | 9,10-ANQ | 208.0572 | 5.6 | 3.2 |
| 1,8-Naphthalic anhydride/ $C_{12}H_6O_3$ | 1,8-NANY | 198.0436 | 6.4 | 5.6 |
| Phenanthrene-9-carboxaldehyde/$C_{15}H_{10}O$ | PHCA | 206.0777 | 5.4 | 4.9 |
| Benzo[a]fluorenone/ $C_{17}H_{10}O$ | BaFLU | 230.0791 | 6.4 | 3.2 |
| 7H-Benz[de]anthracene-7-one/ $C_{17}H_{10}O$ | BANTone | 230.0781 | 7.2 | 5.8 |
| 1-Pyrenecaboxaldehyde/ $C_{17}H_{10}O$ | 1-PYRCA | 230.0786 | 7.5 | 7.2 |
| 1,2-Benzanthraquinone/ $C_{18}H_{10}O_2$ | 1,2-BANQ | 258.0743 | 8.5 | 7.4 |
| 9 NPAHs | | Monitored ions in NCI mode | | |
| 1-Nitronaphthalene/$C_{10}H_7NO_2$ | 1-NNAP | 173.0551 | 4.7 | 4.4 |
| 3-Nitrodibenzofuran/ $C_{12}H_7NO_3$ | 3-NDBF | 213.0475 | 4.4 | 5.1 |
| 5-Nitroacenaphthene/ $C_{12}H_9NO_2$ | 5-NAC | 199.0682 | 5.6 | 5.3 |
| 2-Nitrofluorene/$C_{13}H_9NO_2$ | 2-NFLU | 211.0689 | 5.0 | 5.4 |
| 9-Nitroanthracene/$C_{14}H_9NO_2$ | 9-NANT | 223.0697 | 5.9 | 3.9 |
| 3-Nitrofluoranthene/$C_{16}H_9NO_2$ | 3-NFLT | 247.0688 | 6.4 | 4.1 |
| 1-Nitropyrene/$C_{16}H_9NO_2$ | 1-NPYR | 247.0691 | 3.9 | 3.2 |
| 6-Nitrochrysene/$C_{18}H_{11}NO_2$ | 6-NCHR | 273.0847 | 4.7 | 5.4 |

| | | | | |
|---|---|---|---|---|
| 6-Nitrobenzo[a]pyerene/ $C_{20}H_{11}NO_2$ | 6-NBaP | 297.0845 | 8.4 | 9.7 |


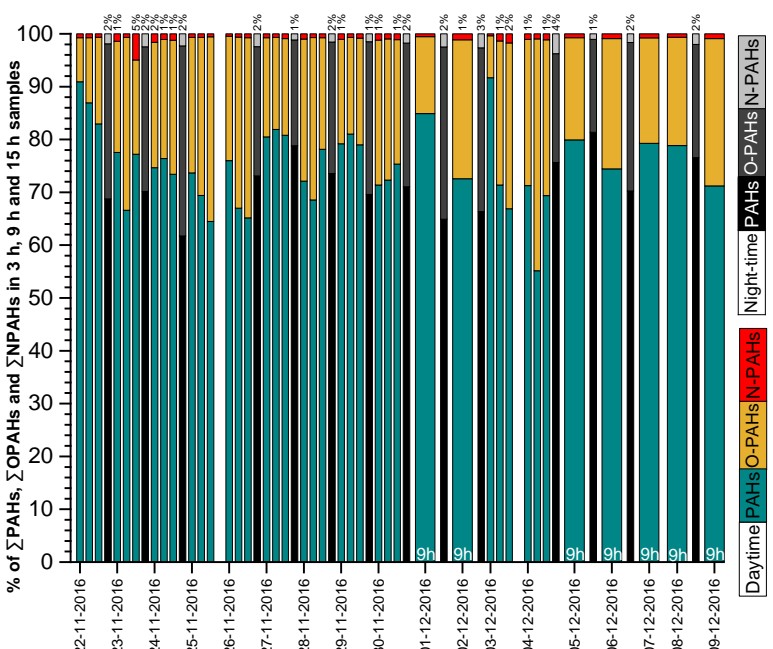

**Figure 2.** Time-series of $\sum$PAHs, $\sum$OPAHs and $\sum$NPAHs in PM$_{2.5}$ samples during the daytime (3 h and 9
h) and night-time (15h). Percentage below 1% for NPAHs are omitted for clarity. Night-time data on
25/11/2016, 03/12/2016 and 07/12/2016 are not available due to lack of samples.



Commented [A52]: Response to Referee 2: Specific comments:
L.672
And Short Comment (SC1)



**Table 2. Minimum, maximum and average atmospheric concentrations of PAHs, OPAHs and NPAHs in PM$_{2.5}$.**
**Compounds in bold represent the highest mean contribution to the sum of all compounds.**

| Compound | Concentrations (ng m$^{-3}$) | | Average contribution to total (%) |
| --- | --- | --- | --- |
| | Minimum-maximum | Average | |
| 16 PAHs | (3 h)/(9 h)/(15 h) | (3 h ± SD*)/(9 h ± SD)/(15 h ± SD) | (3 h)/(9 h)/(15 h) |
| NAP | (0.05-0.8)/(0.1-0.4)/(0.06-0.6) | (0.31 ± 0.2)/(0.24 ± 0.1)/(0.27 ± 0.2) | (0.48)/(0.39)/(0.25) |
| ACY | (0.01-1.2)/(0.1-0.8)/(0.1-1.2) | (0.31 ± 0.3)/(0.31 ± 0.2)/(0.58 ± 0.4) | (0.35)/(0.46)/(0.50) |
| AC | (0.03-0.13)/(0.02-0.09)/(0.01-0.2) | (0.07 ± 0.02)/(0.04 ± 0.03)/(0.07 ± 0.07) | (0.15)/(0.08)/(0.06) |
| FLU | (0.05-1.3)/(0.1-1.0)/(0.1-1.5) | (0.43 ± 0.3)/(0.41 ± 0.3)/(0.63 ± 0.4) | (0.53)/(0.65)/(0.56) |
| **PHE** | **(1.2-23.1)/(1.9-16.3)/(1.5-13.7)** | **(7.38 ± 5.5)/(6.30 ± 5.3)/(8.4 ± 4.0)** | **(8.83)/(9.04)/(7.84)** |
| ANT | (0.5-3.4)/(0.3-1.9)/(0.3-2.9) | (1.07 ± 0.7)/(0.79 ± 0.6)/(1.43 ± 0.8) | (1.48)/(1.23)/(1.32) |
| **FLT** | **(1.4-41.8)/(3.0-17.6)/(3.2-11.7)** | **(12.9 ± 10.0)/(9.10 ± 5.7)/(8.97 ± 2.7)** | **(13.9)/(13.9)/(9.67)** |
| **PYR** | **(0.7-34.6)/(2.1-15.7)/(2.9-10.7)** | **(9.85 ± 8.2)/(7.48 ± 5.0)/(8.1 ± 2.5)** | **(10.4)/(11)/(8.69)** |
| **BaA** | **(1.3-27.7)/(1.3-17.5)/(2.1-18.8)** | **(6.69 ± 6.4)/(6.52 ± 5.9)/(12 ± 5.9)** | **(7.17)/(8.23)/(11)** |
| **CHR** | **(1.4-37.5)/(2.1-20.8)/(2.7-15.9)** | **(10.5 ± 8.7)/(9.17 ± 7.17)/(11.3 ± 4.8)** | **(11.2)/(12.4)/(10.8)** |
| **BbF** | **(1.5-35.3)/(2.1-21.3)/(2.3-20.4)** | **(10.3 ± 8.5)/(8.93 ± 7.3)/(10.8 ± 5.0)** | **(11.2)/(11.8)/(10.4)** |
| BkF | (1.6-15.4)/(1.2-7.4)/(1.3-6.6) | (5.51 ± 3.8)/(3.94 ± 2.7)/(4.3 ± 1.5) | (6.76)/(5.80)/(4.43) |
| **BaP** | **(1.4-37.3)/(1.5-20.7)/(3.2-35.2)** | **(8.81 ± 8.6)/(8.40 ± 7.5)/(18.9 ± 12.4)** | **(9.12)/(10.3)/(16.1)** |
| IcdP | (1.7-16.1)/(0.9-11.6)/(1.0-18.3) | (4.79 ± 3.5)/(4.65 ± 4.3)/(9.75 ± 6.4) | (6.06)/(5.70)/(8.03) |
| DahA | (1.9-5.2)/(0.7-2.9)/(0.5-6.9) | (2.54 ± 0.7)/(1.46 ± 0.8)/(3.0 ± 2.1) | (4.43)/(2.46)/(2.56) |
| BghiP | (2.53-17.0)/(1.2-10.7)/(1.4-15.4) | (5.80 ± 3.5)/(4.70 ± 3.7)/(8.8 ± 5.1) | (7.86)/(6.47)/(7.62) |
| Total | (18-297)/(19-167)/(23-165) | (87.3 ± 58)/(72.5 ± 56)/(107 ± 51) | |
| 10 OPAHs | (3 h)/(9 h)/(15 h) | (3 h ± SD)/(9 h ± SD)/(15 h ± SD) | (3 h)/(9 h)/(15 h) |
| 1,4-NAQ | (0.02-8.1)/(0.16-3.1)/(0.1-4.2) | (2.25 ± 2.4)/(1.27 ± 1.2)/(1.66 ± 1.3) | (6.22)/(5.39)/(3.70) |
| 1-NALD | (0.2-0.8)/(0.07-0.5)/(0.08-0.9) | (0.43 ± 0.1)/(0.20 ± 0.1)/(0.49 ± 0.3) | (2.71)/(1.19)/(1.25) |
| **9-FLON** | **(0.49-14.9)/(0.7-6.0)/(0.8-11.4)** | **(6.76 ± 4.4)/(2.56 ± 1.9)/(4.26 ± 2.8)** | **(25.8)/(14.3)/(10.2)** |
| **9,10-ANQ** | **(0.3-36.4)/(1.2-24.8)/(2.8-36)** | **(8.31 ± 8.8)/(8.65 ± 8.5)/(14.3 ± 9.9)** | **(24.3)/(35.8)/(32.7)** |
| **1,8-NANY[a]** | **(0.3-16.3)/(1.0-6.9)/(3.7-9.3)** | **(7.09 ± 5.4)/(3.69 ± 2.9)/(6.81 ± 2.8)** | **(37.9)/(33.2)/(45.6)** |
| PHCA | (0.1-0.9)/(0.05-0.6)/(0.06-1.9) | (0.26 ± 0.17)/(0.20 ± 0.19)/(0.71 ± 0.57) | (1.42)/(0.99)/(1.50) |
| BaFLU | (0.06-10.8)/(0.1-8.1)/(0.4-15.1) | (2.77 ± 3.0)/(2.72 ± 2.9)/(5.99 ± 4.8) | (7.47)/(9.73)/(12.1) |
| BANTone | (0.08-15.1)/(0.04-8.3)/(0.5-19.8) | (2.46 ± 3.3)/(2.63 ± 2.9)/(9.27 ± 7.3) | (6.10)/(9.05)/(19.1) |
| 1-PYRCA | (0.007-1.8)/(0.008-1.5)/(0.05-2.4) | (0.31 ± 0.4)/(0.39 ± 0.5)/(1.0 ± 0.9) | (0.74)/(1.24)/(1.96) |
| 1,2-BANQ | (0.02-3.6)/(0.03-2.6)/(0.2-3.9) | (0.87 ± 0.96)/(0.90 ± 0.96)/(1.99 ± 1.4) | (2.3)/(3.24)/(4.33) |

**Commented [A53]:** Response to Referee 2: Comment 1

| | | | |
|---|---|---|---|
| Total | (1.8-87.9)/(3.6-55.3)(8.9-95.5) | (28.7 ± 21)/(21.7 ± 18)/(41.6 ± 26) | |
| 9 NPAHs | (3 h)/(9 h)/(15 h) | (3 h ± SD)/(9 h ± SD)/(15 h ± SD) | (3 h)/(9 h)/(15 h) |
| 1-NNAP | (0.01-0.1)/(0.008-0.04)/(0.005-0.03) | (0.03 ± 0.02)/(0.01 ± 0.01)/(0.01 ± 0.008) | (4.38)/(3.08)/(0.57) |
| **3-NDBF** | **(0.08-1.5)/(0.02-0.06)/(0.03-2.4)** | **(0.33 ± 0.31)/(0.03 ± 0.01)/(0.89 ± 0.84)** | **(33.4)/(7.92)/(22.4)** |
| 5-NAC[b] | (0.04-0.1)/(<LOQ)/(0.03-0.35) | (0.08 ± 0.05)/(<LOQ )/(0.18 ± 0.13) | (5.64)/(<LOQ )/(4.67) |
| 2-NFLU | (0.03-0.3)/(0.01-0.3)/(0.01-0.5) | (0.08 ± 0.06)/(0.09 ± 0.11)/(0.26 ± 0.21) | (10.15)/(10.00)/(7.28) |
| **9-NANT** | **(0.01-1.2)/(0.06-0.1)/(0.4-2.4)** | **(0.36 ± 0.37)/(0.41 ± 0.31)/(1.18 ± 0.6)** | **(27.1)/(53.4)/(47.5)** |
| **3-NFLT** | **(0.05-1.2)/(0.02-0.5)/(0.04-1.2)** | **(0.34 ± 0.3)/(0.21 ± 0.2)/(0.54 ± 0.4)** | **(24.6)/(23.4)/(17.7)** |
| 1-NPYR[c] | (0.01-0.1)/(0.01-0.06)/(0.008-0.2) | (0.05 ± 0.03)/(0.06 ± 0.05)/(0.02 ± 0.02) | (2.92)/(2.48)/(2.01) |
| 6-NCHR[d] | (0.05-0.2)/(<LOQ)/(0.009-0.02) | (0.09 ± 0.06)/(<LOQ )/(0.01 ± 0.007) | (5.6)/(<LOQ )/(0.5) |
| 6-NBaP[e] | (<LOQ )/(<LOQ)/(0.02-0.08) | (<LOQ )/(<LOQ)/(0.05 ± 0.01) | (<LOQ )/(<LOQ)/(1.26) |
| Total | (0.15-3.92)/(0.13-2.0)/(0.57-6.43) | (1.17 ± 1.0)/(0.80 ± 0.66)/(3.06 ± 1.8) | |

[a] Quantified in 28/54 samples
[b] Quantified in 7/54 samples
[c] Quantified in 35/54 samples
[d] Quantified in 5/54 samples
[e] Quantified in 11/54 samples
* SD: Standard Deviation

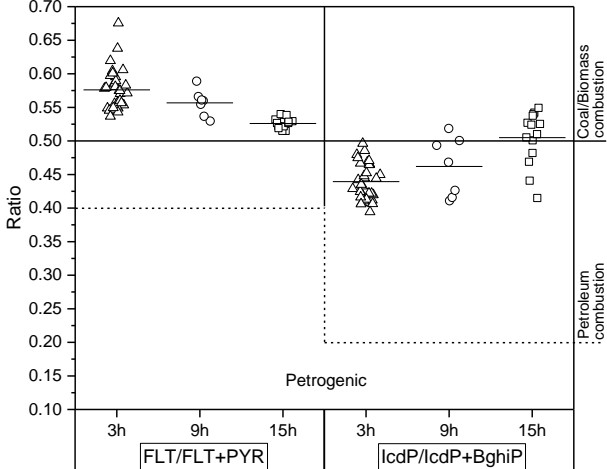

**Figure 3. Column scatter of FLT/(FLT + PYR) and IcdP/(IcdP + BghiP) in the particulate phase at three different time sampling averages, open triangles and circles represent the daytime data for 3 h and 9 h samples, open squares represent the night-time data of 15 h. The dashed line separates the petroleum combustion source from petrogenic source for both ratios. The solid short line on each data set represent the mean value of ratios.**

**Table 3. Average concentration of $\sum BaP_{eq}$ in ng m$^{-3}$ and cancer risk assessment for the sum of 16PAHs, 1OPAH and 6NPAHs.**

| Sampling hours | $\sum[BaP]_{eq}$ ng m$^{-3}$ | $UR_{BaP} = 1.1 \times 10^{-6}$ (CalEPA) | $UR_{BaP} = 8.7 \times 10^{-5}$ (WHO) | Risk per million people |
|---|---|---|---|---|
| 9 h (daytime, n=40)[a] | 15.9[a] | 1.75 x 10$^{-5}$ | 1.38 x 10$^{-3}$ | 17[b] – 1383[c] |
| 15 h (night-time, n=14) | 28.3 | 3.17 x 10$^{-5}$ | 2.46 x 10$^{-3}$ | 31[b] – 2460[c] |
| **24 h (n=54)** | **23.6** | **2.6 x 10$^{-5}$** | **2.05 x 10$^{-3}$** | **26[b] – 2053[c]** |

[a] Average includes combined 3 h samples in each day (n=33) and 9 h samples (n=7)
[b] Calculated Value according to CalEPA
[c] Calculated Value according to WHO
n: number of samples


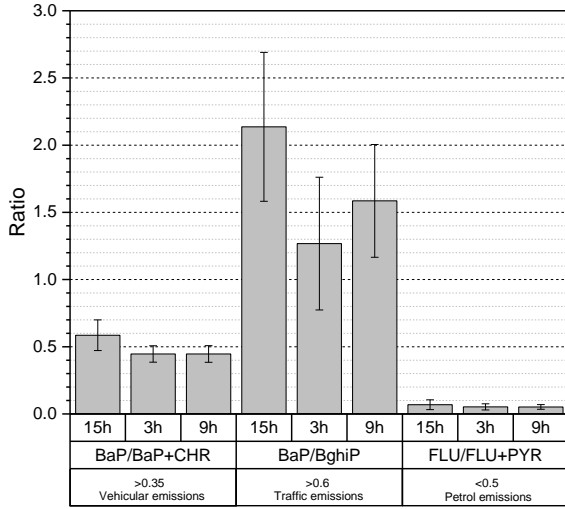

**Figure 4. Column distribution of BaP/BaP+CHR, BaP/BghiP and FLU/FLU+PYR in the particulate phase for three**
**different sampling periods. 3 h and 9 h represent samples collected during the day and 15 h for samples at night.**
**Error bars reflect standard deviations.**


















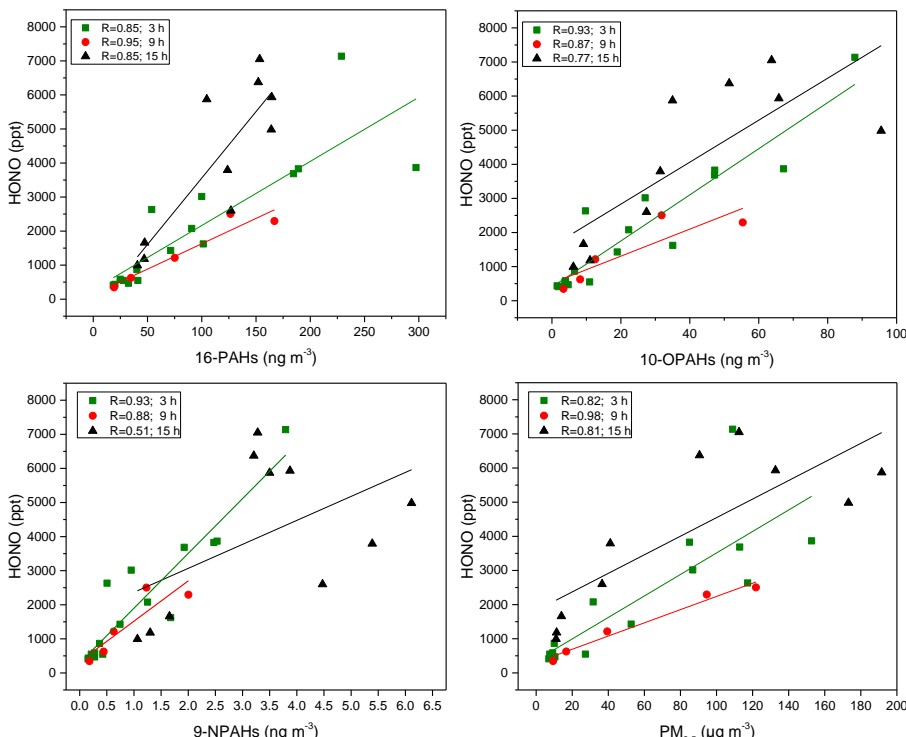

**Figure 5. Correlation coefficients of ∑PAHs, ∑OPAHs, ∑NPAHs and PM₂.₅ with HONO. Time sampling resolution of**
**3 h and 9 h refer to daytime concentrations and 15 h to nocturnal concentrations. Significance levels were between**
**0.001 and 0.05 except for HONO and NPAHs at night, P level > 0.05 and Pearson coefficient 0.52.**



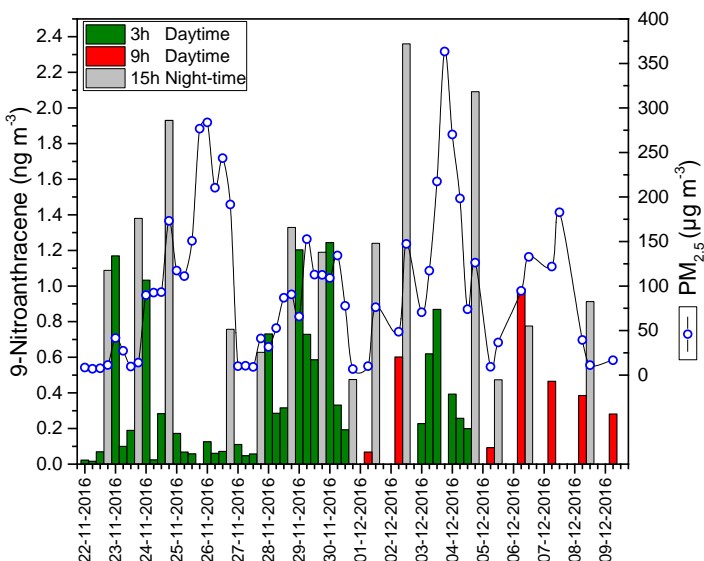


**Figure 6.** Temporal variation of 9-Nitroanthrance and PM2.5 over the entire winter campaign.