# Peer review of "Variability of polycyclic aromatic hydrocarbons and their"

_Atmospheric Chemistry and Physics, 2019_

## Short Comment (SC1) · 6 Mar 2019

[revised manuscript text omitted]

---

## Referee Comment (RC1) · Anonymous Referee #1 · 14 Mar 2019

This is a nicely written paper that contributes new data to help improve understanding of urban sources and levels of PAHs and their derivatives, including day vs night variability. What is still missing from the paper in my opinion, is an expression of the study's significance. i.e. what is novel about this study and how/where do the results make an impact on advancing the field of science. I would expect this for a paper in ACP.

Minor things:

Line 76-86: Why only target PM and not gas-phase? Some explanation is warranted here I think. Also, when comparing results to other studies is this always based on PM and not total?

Line 83: Do the longer nighttime samples compared to daytime samples introduce a source of bias in the results? For instance, related to particle capture efficiency which changes as the filter becomes more loaded. Another question is heterogeneous reactions on the filter during the 15hrs the samples are being collected and exposed to a high volume of atmospheric oxidants. If these reactions are occurring they will impact the 15h samples to a greater extent. This issue has been considered in the literature and I think the authors should at least raise this concern.

Line 203-205 and elsewhere: please use consistent number of significant figures when reporting concentrations. In this section it ranges from 2 to 4. I think that either 2 or 3 significant figures is appropriate given method uncertainties.

Line 369+: I think the spatial variability within Beijing may be even more important than seasonal variability. The authors believe their results are representative of "Beijing" and present them as e.g. "...concentrations for Beijing". However, I wonder if that is appropriate without first investigating the representativeness of their sampling site for the entire metropolis of Beijing. There is also the potential for a dominant impact due to a local source (or sources) that may impact or bias results at a given site. The authors should be careful to address this possibility and not overstate the representativeness of their results (or other results from single sites in other studies), when making comparisons e.g. Beijing vs Athens. end/

---

## Author Comment (AC1) · 9 Apr 2019

**Reply to Alexandre Albinet comments**
Author's response in Blue

**Line 42-43:** Cite : Ravindra, K., Sokhi, R. and Van Grieken, R.: Atmospheric polycyclic aromatic hydrocarbons: Source attribution, emission factors and regulation, Atmospheric Environment, 42, 2895–2921, doi:doi: DOI: 10.1016/j.atmosenv.2007.12.010, 2008.
Reference added

**Line 52:** Ringuet, J., Albinet, A., Leoz-Garziandia, E., Budzinski, H. and Villenave, E.: Reactivity of polycyclic aromatic compounds (PAHs, NPAHs and OPAHs) adsorbed on natural aerosol particles exposed to atmospheric oxidants, Atmospheric Environment, 61, 15–22, doi:10.1016/j.atmosenv.2012.07.025, 2012.
Reference added

**Line 53-54:**
-Nalin, F., Golly, B., Besombes, J.-L., Pelletier, C., Aujay-Plouzeau, R., Verlhac, S., Dermigny, A., Fievet, A., Karoski, N., Dubois, P., Collet, S., Favez, O. and Albinet, A.: Fast oxidation processes from emission to ambient air introduction of aerosol emitted by residential log wood stoves, Atmospheric Environment, 143, 15–26, doi:10.1016/j.atmosenv.2016.08.002, 2016.
-W.F. Rogge, L.M. Hildemann, M.A. Mazurek, G.R. Cass, B.R.T. Simoneit, Sources of fine organic aerosol. 2. Noncatalyst and catalyst-equipped automobiles and heavy-duty diesel trucks Environ. Sci. Technol., 27 (1993), pp. 636-651
-B. Zielinska, J. Sagebiel, J.D. McDonald, K. Whitney, D.R. Lawson. Emission rates and comparative chemical composition from selected in-use diesel and gasoline-fueled Vehicles J. Air Waste Manag. Assoc., 54 (2004), pp. 1138-1150
-W.F. Rogge, L.M. Hildemann, M.A. Mazurek, G.R. Cass, B.R.T. Simoneit Sources of fine organic aerosol. 9. Pine, oak and synthetic log combustion in residential fireplaces Environ. Sci. Technol., 32 (1998), pp. 13-2
Two references added (Nalin et al., 2016; Rogge et al., 1993)

**Line 61:** Tomaz, S., Shahpoury, P., Jaffrezo, J.-L., Lammel, G., Perraudin, E., Villenave, E. and Albinet, A.: One-year study of polycyclic aromatic compounds at an urban site in Grenoble (France): Seasonal variations, gas/particle partitioning and cancer risk estimation, Science of The Total Environment, 565, 1071–1083, doi:10.1016/j.scitotenv.2016.05.137, 2016.
Reference added

**Line 62:**
Please explain which limitations. Sampling artifacts, etc...
Have a look there and references included:
-Albinet, A., Leoz-Garziandia, E., Budzinski, H. and Villenave, E.: Sampling precautions for the measurement of nitrated polycyclic aromatic hydrocarbons in ambient air, Atmospheric Environment, 41(23), 4988–4994, doi:16/j.atmosenv.2007.01.061, 2007.
-Albinet, A., Papaiconomou, N., Estager, J., Suptil, J. and Besombes, J.-L.: A new ozone denuder for aerosol sampling based on an ionic liquid coating, Analytical and Bioanalytical Chemistry, 396, 857–864, doi:10.1007/s00216-009-3243-5, 2009.
-Goriaux, M., Jourdain, B., Temime, B., Besombes, J.-L., Marchand, N., Albinet, A., Leoz-Garziandia, E. and Wortham, H.: Field Comparison of Particulate PAH Measurements Using a Low-Flow Denuder Device and Conventional Sampling Systems, Environ. Sci. Technol., 40(20), 6398–6404, doi:10.1021/es060544m, 2006.

References added and text corrected. More details added to ''error evaluation'' please see below author's comments on Line 156.

Author's changes in manuscript (L.62).
"However, a long sampling and averaging period creates some limitations such as sampling artefacts, notably where changing atmospheric photolysis conditions (air humidity, temperature, wind direction, ozone or other oxidant concentrations) may have a significant influence on PAHs concentrations and oxidation rates (Albinet et al., 2007b; Albinet et al., 2009, Goriaux et al., 2006, Tsapakis and Stephanou., 2003; Tsapakis and Stephanou., 2007, Ringuet et al., 2012b)."

**Line 67:** please cite also: Srivastava, D., Favez, O., Bonnaire, N., Lucarelli, F., Haeffelin, M., Perraudin, E., Gros, V., Villenave, E. and Albinet, A.: Speciation of organic fractions does matter for aerosol source apportionment. Part 2: Intensive short-term campaign in the Paris area (France), Science of The Total Environment, 634, 267–278, doi:10.1016/j.scitotenv.2018.03.296, 2018.
Reference added

**Line 67-69:** and other ones such as:
-Reisen, F. and Arey, J.: Atmospheric reactions influence seasonal PAH and nitro-PAH concentrations in the Los Angeles basin, Environ. Sci. Technol., 39(1), 64–73, doi:10.1021/es035454l, 2004.
-Albinet, A., Leoz-Garziandia, E., Budzinski, H. and ViIlenave, E.: Polycyclic aromatic hydrocarbons (PAHs), nitrated PAHs and oxygenated PAHs in ambient air of the Marseilles area (South of France): Concentrations and sources, Science of The Total Environment, 384(1-3), 280–292, doi:10.1016/j.scitotenv.2007.04.028, 2007.
-Albinet, A., Leoz-Garziandia, E., Budzinski, H., Villenave, E. and Jaffrezo, J.-L.: Nitrated and oxygenated derivatives of polycyclic aromatic hydrocarbons in the ambient air of two French alpine valleys: Part 1: Concentrations, sources and gas/particle partitioning, Atmospheric Environment, 42(1), 43–54, doi:10.1016/j.atmosenv.2007.10.009, 2008.
-Srivastava, D., Favez, O., Bonnaire, N., Lucarelli, F., Haeffelin, M., Perraudin, E., Gros, V., Villenave, E. and Albinet, A.: Speciation of organic fractions does matter for aerosol source apportionment. Part 2: Intensive short-term campaign in the Paris area (France), Science of The Total Environment, 634, 267–278, doi:10.1016/j.scitotenv.2018.03.296, 2018.
Three references added (Albinet et al., 2008; Reisen and Arey., 2004; Srivastava et al., 2018.) text modified.
Author's changes in manuscript (L.67).
A few studies have used twice daily (12 h) sampling (Albinet et al., 2008; Zhang et al., 2018; Farren et al., 2015; Ringuet et al., 2012b), obtaining limited information on variability in concentrations during the daytime and night-time (Tsapakis and Stephanou., 2007). Shorter time periods for sampling (3 h and 4 h) are still very limited (Reisen and Arey., 2004; Srivastava et al., 2018).

**Line 99:** clean I guess
and that's not only the purpose, see:
-Albinet, A., Leoz-Garziandia, E., Budzinski, H. and ViIlenave, E.: Simultaneous analysis of oxygenated and nitrated polycyclic aromatic hydrocarbons on standard reference material 1649a (urban dust) and on natural ambient air samples by gas chromatography-mass spectrometry with negative ion chemical ionisation, Journal of Chromatography A, 1121(1), 106–113, doi:16/j.chroma.2006.04.043, 2006.
-Albinet, A., Nalin, F., Tomaz, S., Beaumont, J. and Lestremau, F.: A simple QuEChERS-like extraction approach for molecular chemical characterization of organic aerosols: application to nitrated and oxygenated PAH derivatives (NPAH and OPAH) quantified by GC–NICIMS, Anal Bioanal Chem, 406(13), 3131–3148, doi:10.1007/s00216-014-7760-5, 2014.
Author's changes in manuscript (L.99).

All samples and blanks were purified on a SPE silica normal phase cartridge (1g/6ml; SIGMA ALDRICH) to reduce the impacts of interfering compounds in the matrix and to help maintain a clean GC injection inlet liner.

**Line 126:** You can say that it's a modified version of previous published methods:
-Albinet, A., Leoz-Garziandia, E., Budzinski, H. and Villenave, E.: Simultaneous analysis of oxygenated and nitrated polycyclic aromatic hydrocarbons on standard reference material 1649a (urban dust) and on natural ambient air samples by gas chromatography-mass spectrometry with negative ion chemical ionisation, Journal of Chromatography A, 1121(1), 106–113, doi:16/j.chroma.2006.04.043, 2006.
-Albinet, A., Nalin, F., Tomaz, S., Beaumont, J. and Lestremau, F.: A simple QuEChERS-like extraction approach for molecular chemical characterization of organic aerosols: application to nitrated and oxygenated PAH derivatives (NPAH and OPAH) quantified by GC–NICIMS, Anal Bioanal Chem, 406(13), 3131–3148, doi:10.1007/s00216-014-7760-5, 2014.
-Kawanaka Y, Sakamoto K, Wang N, Yun S-J (2007) Simple and sensitive method for determination of nitrated polycyclic aromatic hydrocarbons in diesel exhaust particles by gas chromatography-negative ion chemical ionisation tandem mass spectrometry. J Chromatogr A 1163:312–317
-Bezabeh DZ, Bamford HA, Schantz MM, Wise SA (2003) Determination of nitrated polycyclic aromatic hydrocarbons in diesel particulate-related standard reference materials by using gas chromatography/mass spectrometry with negative ion chemical ionization. Anal Bioanal Chem 375:381–388
References added and text corrected
Author's changes in manuscript (L.126).
The method development for OPAHs and NPAHs was based on previous studies (Albinet et al., 2006; Albinet et al., 2014; Bezabeh et al., 2003; Kawanaka et al., 2007) and conducted using Negative Chemical Ionisation

**Line 126:** 155 eV, really?
Yes 155 ev (autotune settings), According to Agilent manual the maximum can be used in NCI is 240 ev.

**Line 156-157:**
Clearly I doubt about it. If you really want to consider all the possible errors, you should applied a GUM approach. https://sisu.ut.ee/measurement/9-iso-gum-modeling-approach-bottom-approach
In addition, to estimate the uncertainties, one of the main factor is linked to the extraction efficiency and you don't mention anything about that. Standard reference materials are useful for that (for both PAHs and PAH derivatives). There are many paper about that:
-Albinet, A., Leoz-Garziandia, E., Budzinski, H. and Villenave, E.: Simultaneous analysis of oxygenated and nitrated polycyclic aromatic hydrocarbons on standard reference material 1649a (urban dust) and on natural ambient air samples by gas chromatography-mass spectrometry with negative ion chemical ionisation, Journal of Chromatography A, 1121(1), 106–113, doi:16/j.chroma.2006.04.043, 2006.
-Albinet, A., Nalin, F., Tomaz, S., Beaumont, J. and Lestremau, F.: A simple QuEChERS-like extraction approach for molecular chemical characterization of organic aerosols: application to nitrated and oxygenated PAH derivatives (NPAH and OPAH) quantified by GC–NICIMS, Anal Bioanal Chem, 406(13), 3131–3148, doi:10.1007/s00216-014-7760-5, 2014.
-Schantz MM, McGaw E, Wise SA (2012) Pressurized liquid extraction of diesel and air particulate standard reference materials: effect of extraction temperature and pressure. Anal Chem 84:8222–8231
-Wise S, Poster D, Kucklick J, Keller J, VanderPol S, Sander L, Schantz M (2006) Standard reference materials (SRMs) for determination of organic contaminants in environmental samples. Anal Bioanal Chem 386:1153–1190

-Delgado-Saborit JM, Alam MS, Godri Pollitt KJ, Stark C, Harrison RM (2013) Analysis of atmospheric concentrations of quinones and polycyclic aromatic hydrocarbons in vapour and particulate phases. Atmos Environ 77:974–982

-Nocun M, Schantz M (2013) Determination of selected oxygenated polycyclic aromatic hydrocarbons (oxy-PAHs) in diesel and air particulate matter standard reference materials (SRMs). Anal Bioanal Chem 405:5583–5593

-Ahmed, T. M., Bergvall, C., Åberg, M. and Westerholm, R.: Determination of oxygenated and native polycyclic aromatic hydrocarbons in urban dust and diesel particulate matter standard reference materials using pressurized liquid extraction and LC–GC/MS, Anal Bioanal Chem, 1–12, doi:10.1007/s00216-014-8304-8, 2014.

The estimated error (L.157) is due to available information on laboratory test performance. The word ''uncertainty'' (L.157) was replaced by ''error''.

Based on our type of data we chose the ''top-down'' approach where the bias determination can be based on recovery efficiency. Using the step-by-step approach (bottom up) will increase the uncertainty on individual compound and mostly used for ISO accreditation. It is shown in the text that the average recovery efficiencies have ranged from 85% to 96% (L.102). The %RSD average for deuterium labelled compounds was about 3.6% (L.149). In this study we spiked 10 filters covering the different sampling time (3h, 9h, 15h). We agree that SRM are widely used for a better assessment of the analytical method and their use is probably required for publishing in analytical technique Journals. SRM do not provide certified values for most of the derivatives compounds we have used.

Author's changes in manuscript (L.156).

Therefore, the overall estimated error, combining the precision and the systematic effects, is less than 20%.

Another source of error can be attributed to sampling artefacts and this has been discussed in previous studies (Schauer, C. et al., 2003, Goriaux, M. et al., 2006, Tsapakis and Stephanou, 2003). The absence of an ozone denuder to trap the gas phase oxidants may lead to an underestimation of the true values of PAHs due to chemical decomposition. Therefore, data from long sampling times and under high ozone ambient concentrations may be biased by sampling artefacts by more than 100 % (Schauer et al., 2003, Goriaux et al., 2006). However, at low ozone levels, negative artefacts were considered not significant (Tsapakis and Stephanou 2003), whilst, at medium ozone levels (30-50 ppb), PAHs values were underestimated by 30 % (Schauer et al., 2003). In addition, heterogeneous reactions during particles sampling may occur on the monolayer surface coverage only with limited diffusion of oxidants to the bulk particles (Keyte et al., 2013 and references therein). Previous studies reported that the formation of NPAHs during high-volume sampling is not significant and calculated to be < 3 % (Arey et al., 1988) and < 0.1 % (Dimashki et al., 2000).

Considering the predominant role of ozone levels (below 30 ppb in this study, mean: 8.4 ppb), sampling time and temperature on the magnitude of PAHs concentrations, the estimation of the negative sampling artefacts on our data range between 10 and 20 %, with the highest error estimation attributable to longer sampling time (15h).

**Line 159:** lab or field blanks? How many blanks?
**Line 160:** if they are detected they are not <LOD
**Line 161:** how much?
Text corrected
Author's changes in manuscript (L.159,160,161).

To determine any sources of contamination during sample preparation and the analytical procedure, the solvent (acetonitrile) and field blanks (n=2) were analysed following the same procedure as for the samples (Extraction, SPE, Evaporation). Most target compounds were found to be below LOD (S/N=3) or orders of magnitude (up to $10^3$- $10^4$) lower than was found in the samples. A small number of compounds found in field blanks (1,8-Naphthalic anhydride, Benzo[a]fluorenone, 1-Nitronaphthalene, 9-Nitroanthracene) have a higher contribution (4-30 %) to very few filters (2 to 5 samples) collected

over a 3 h time period, if this was co-incident with low particulate loading conditions. The contribution to each compound from field blanks has been corrected in the final data.

**Line 218:** All this part should be moderated or removed.
There are clearly evidences that the validity of this kind of ratios is limited
-Dvorská, A., Lammel, G. and Klánová, J.: Use of diagnostic ratios for studying source apportionment and reactivity of ambient polycyclic aromatic hydrocarbons over Central Europe, Atmospheric Environment, 45(2), 420–427, doi:10.1016/j.atmosenv.2010.09.063, 2011.
-Dvorská, A., Komprdová, K., Lammel, G., Klánová, J. and Plachá, H.: Polycyclic aromatic hydrocarbons in background air in central Europe – Seasonal levels and limitations for source apportionment, Atmospheric Environment, 46, 147–154, doi:10.1016/j.atmosenv.2011.10.007, 2012.
-Katsoyiannis, A., Sweetman, A. J. and Jones, K. C.: PAH molecular diagnostic ratios applied to atmospheric sources: a critical evaluation using two decades of source inventory and air concentration data from the UK, Environ. Sci. Technol., 45(20), 8897–8906, doi:10.1021/es202277u, 2011.
-Katsoyiannis, A. and Breivik, K.: Model-based evaluation of the use of polycyclic aromatic hydrocarbons molecular diagnostic ratios as a source identification tool, Environmental Pollution, 184, 488–494, doi:10.1016/j.envpol.2013.09.028, 2014.
Limitations for source apportionment using Diagnostic Ratios (DR) increase when the receptor site is far from evident emission sources, e.g. long-time sampling (24h) or by using volatile and reactive compounds such as the ratio ANT/(ANT+PHE) which indicate petrogenic sources. This ratio has not been used in this study and the sampling time scale was short (3h during daytime) which reduce limitations. In addition, the site location (Fig S1) was at urban area and surrounded by busy traffic road, residential buildings, underground, restaurants and thermal power stations.
A recent study (Feng et al., 2019) has used PMF models and DR and showed consistent results from both methods.
Accordingly, we consider the obtained results from diagnostic ratios are representative and may be important for future studies and method comparison.
Reference:
Feng, B., Li, L., Xu, H., Wang, T., Wu, R., Chen, J., Zhang, Y., Liu, S., Ho, S.S.H., Cao, J., Huang, W.: $PM_{2.5}$-bound polycyclic aromatic hydrocarbons (PAHs) in Beijing: Seasonal variations, sources, and risk assessment, Journal. of Environmental. Sciences., 77, 11-19, doi:10.1016/j.jes.2017.12.025, 2019.

**Line 246** please cite actual references which showed that, e.g.
-I.J. Keyte, A. Albinet, R.M. Harrison.On-road traffic emissions of polycyclic aromatic hydrocarbons and their oxy- and nitro- derivative compounds measured in road tunnel environments.Sci. Total Environ., 566–567 (2016), pp. 1131-1142
-J.K. Schulte, J.R. Fox, A.P. Oron, T.V. Larson, C.D. Simpson, M. Paulsen, N. Beaudet, J.D. Kaufman, S. Magzamen. Neighborhood-scale spatial models of diesel exhaust concentration profile using 1-nitropyrene and other nitroarenes.Environ. Sci. Technol., 49 (2015), pp. 13422-13430
-B. Zielinska, J. Sagebiel, W.P. Arnott, C.F. Rogers, K.E. Kelly, D.A. Wagner, J.S. Lighty, A.F. Sarofim, G. Palmer Phase and size distribution of polycyclic aromatic hydrocarbons in diesel and gasoline vehicle emissions Environ. Sci. Technol., 38 (2004), pp. 2557-2567
-B. Zielinska, J. Sagebiel, J.D. McDonald, K. Whitney, D.R. Lawson. Emission rates and comparative chemical composition from selected in-use diesel and gasoline-fueled vehicles. J. Air Waste Manage. Assoc., 54 (2004), pp. 1138-1150.
Two references added (Keyte et al., 2016, Magzamen et al., 2015)

**Line 252:** Tomaz, S., Jaffrezo, J.-L., Favez, O., Perraudin, E., Villenave, E. and Albinet, A.: Sources and atmospheric chemistry of oxy- and nitro-PAHs in the ambient air of Grenoble (France), Atmospheric Environment, 161, 144–154, doi:10.1016/j.atmosenv.2017.04.042, 2017.
Reference added

**Line 312-313:** OK but ANT is mainly in the gas phase and you have data only about the PM phase

As it was shown in the text (L. 299-300) the dominant formation pathway of NPAHs is secondary formation in the gas phase. Heterogeneous reactions may play a role and contribute to the formation of NPAHs. 9-nitroanhtracene was reported to be one of the major products for the heterogeneous reaction of adsorbed anthracene on different types of particles (NaCl, $SiO_2$, MgO) in presence of $NO_2$ (Jinzhu et al., 2011; Wenyuan and Tong., 2014).

Zhang et al, (2013) have reported that ANT in the gas phase can be adsorbed on particles and they showed a high production of 9-NANT during the heterogeneous reaction of adsorbed ANT in presence of $O_3$-NO, suggesting that $NO_3$ radicals were formed and reacted with anthracene. In our study the positive strong correlation between ANT and 9-NANT in particle phase does not prove but it supports the heterogeneous formation pathway of 9-NANT. On the other hand, previous studies reported that the formation of NPAHs during high-volume sampling is not significant and calculated to be < 3% (Arey et al., 1988) and < 0.1 % (Dimashki et al., 2000). The mechanisms and levels of formation are still far from being fully understood, especially that the reaction of PAHs with oxidants is highly influenced by surface coverage and PAHs loading (Keyte, I.J., 2013 and references therein) i.e. particles were shown to exhibit a potential inhibiting factor on the reactivity of PAHs due to slow diffusion of oxidants and inaccessibility of PAHs in the bulk particle.

The gas phase reaction of ANT with $NO_3$ radicals and the formation of 9-NANT either in the gas phase or adsorbed on particles still unclear. Future studies might help us to better understand the chemical transformation of anthracene in the atmosphere, this suggest that probably both pathways contribute to the formation of 9-NANT on particles. Accordingly, this explain why 9-NANT is the most abundant NPAH in this study and in parallel makes ANT one the lowest concentrations in particle phase.

References:
Jinzhu. M. et al., Atmospheric Environment 45, 917-924, 2011.
Wenyuan, C. et al., Environ. Sci. Technol., 48, 8671–8678, doi:10.1021/es501543g, 2014.
Yang, Z. et al., Journal of Environmental Sciences, 25, 1817–1823, 2013.
Arey, J. et al., Env. Sci. Tech, 22, 457-462, 1988
Dimashki, M. et al., Atmospheric Environment 34, 2459-2469, 2000.

**Line 318:** For all this part you should cite the previous articles where this approach has been described and used:

-A. Albinet, E. Leoz-Garziandia, H. Budzinski, E. Villenave, J.-L. Jaffrezo Nitrated and oxygenated derivatives of polycyclic aromatic hydrocarbons in the ambient air of two French alpine valleys: part 1: concentrations, sources and gas/particle partitioning Atmos. Environ., 42 (2008), pp. 43-54,

-B.A.M. Bandowe, H. Meusel, R. Huang, K. Ho, J. Cao, T. Hoffmann, W. Wilcke PM2.5-bound oxygenated PAHs, nitro-PAHs and parent-PAHs from the atmosphere of a Chinese megacity: seasonal variation, sources and cancer risk assessment Sci. Total Environ., 473-474 (2014), pp. 77-87, 10.1016/j.scitotenv.2013.11.108

-W. Huang, B. Huang, X. Bi, Q. Lin, M. Liu, Z. Ren, G. Zhang, X. Wang, G. Sheng, J. Fu. Emission of PAHs, nitro-PAHs and oxy-PAHs, from residential honeycomb coal briquette combustion. Energy Fuel, 28 (2013), pp. 636-642, 10.1021/ef401901d

-Tomaz, S., Shahpoury, P., Jaffrezo, J.-L., Lammel, G., Perraudin, E., Villenave, E. and Albinet, A.: One-year study of polycyclic aromatic compounds at an urban site in Grenoble (France): Seasonal variations, gas/particle partitioning and cancer risk estimation, Science of The Total Environment, 565, 1071–1083, doi:10.1016/j.scitotenv.2016.05.137, 2016.

Two references added (Albinet et al., 2008; Tomaz et al., 2016), Bandowe et al. 2014 already cited.

**Line 321-322:** That are not the references showing that

References moved beside "commonly used"

Author's changes in manuscript (L.321-322)

For PAHs, Benzo[a]pyrene was chosen as the reference chemical because it is known as the most carcinogenic PAH (OEHHA., 1994, 2002) and is commonly used (Albinet et al., 2008; Tomaz et al., 2016; Alves et al., 2017; Bandowe et al., 2014; Ramírez et al., 2011) as an indicator of carcinogenicity of total PAHs.

**Line 336-337:** Please specify in a Table which TEF values have benne used and the references

The toxicity Equivalency Factor (TEFs) and references used for individual PAHs, OPAHs and Nitro-PAHs are shown in supporting information on Table S3.

**Line 375-379:**

-Albinet, A., Leoz-Garziandia, E., Budzinski, H., Villenave, E. and Jaffrezo, J.-L.: Nitrated and oxygenated derivatives of polycyclic aromatic hydrocarbons in the ambient air of two French alpine valleys: Part 1: Concentrations, sources and gas/particle partitioning, Atmospheric Environment, 42(1), 43–54, doi:10.1016/j.atmosenv.2007.10.009, 2008.

-Tomaz, S., Shahpoury, P., Jaffrezo, J.-L., Lammel, G., Perraudin, E., Villenave, E. and Albinet, A.: One-year study of polycyclic aromatic compounds at an urban site in Grenoble (France): Seasonal variations, gas/particle partitioning and cancer risk estimation, Science of The Total Environment, 565, 1071–1083, doi:10.1016/j.scitotenv.2016.05.137, 2016.

Tomaz et al., 2016 cited. Text modified.

Author's changes in manuscript (L.375)

Our results were considerably higher than those estimated for western European cities during the winter, such as Grenoble: 1.4 ng m$^{-3}$ (Tomaz et al., 2016), Oporto: 3.56 ng m$^{-3}$, Florence: 1.39 ng m$^{-3}$ and Athens: 0.43 ng m$^{-3}$ (Alves et al., 2017). ECR values estimated for each city were 31 (Grenoble), 6.6 (Oporto), 17 (Florence) and 54 (Athens) times lower than our ECR estimation.

**Line 672:** Figure 2 Really difficult to read

Slightly modified and re-arranged. Please see below

[Figure]

---

## Referee Comment (RC2) · Anonymous Referee #2 · 15 Apr 2019

The authors present a rich data set examining the concentrations of PAH's in atmospheric particulate matter in Beijing. PAHs are known to increase the toxicity of aerosols and this work advances our understanding of their dynamics in the atmosphere. The analytical methodology applied in this work is impressive and carefully reported. Where this manuscript needs improvement is the discussion and presentation of the data. I recommend the authors work to increase the clarity of the results, and their interpretations, to ensure their work is as impactful as possible.

**General Comments:**

Comment 1: The authors have a very data rich paper, but often don't report a measure of variability (e.g., standard deviation). This is particularly potent given the title of the manuscript. In the text and in the tables a measure of variability should be included, and discussed, when reporting any data. Specifically, standard deviations should be included in table 2.

Comment 2: The authors present this work as a higher resolution examination of PAH dynamics because they use relatively high resolution (3h) samples. Despite this, there is almost no discussion of temporal dyanmics other than day vs. night or 3h vs. 9h vs. 15 h. I would like to see the authors look into the temporal dynamics on given days and investigate drivers of PAH variability beyond correlation.

Comment 3: Was there meteorological data available at your study site? PAH lifetime, emission and phase are likely influenced by temperature or solar flux. Additionally, analysis of wind direction trends during the specific emission periods would provide additional insight into the importance of different sources. If this data exists, including it in the analysis could increase the impact of the results

Comment 4: Organization. This manuscript, although data rich, is poorly organized with many acronyms, lots of in text data, and unclear conclusions. I recommend the authors make an attempt to trim the text. Specifically, many data references in the discussion could be replaced with reference to the appropriate figure or table. This will allow the readers to better understand the author's conclusions by cleaning up the text.

Comment 5: The authors could improve their argument for the specific significance of their work. The authors do a good job of detailing PAHs in aerosol but don't discuss their site in significant detail. Please elaborate on why there is a pressing need to study PAHs in Beijing within your introduction

Comment 6: For a field study, site selection and significance is very important. Please add some discussion of how representative your site is compared to Beijing in general, what the strengths of the site are, and what are some possible weaknesses.

**Specific Comments**

Line 20: Restructure the list of compounds. As written, it is easy to miss the period after the major PAHs and not realize a new list has started for the major OPAHs. Use the same sentence structure for each type to avoid confusing the reader.

Line 38: comma after "organic carbon"

Line 58: Clarify "geographic peripheral expansion"

Line 60: change "PAH on 24 $PM_{2.5}$ sampling" to "PAH concentrations in 24 h averaged samples".

Line 62: Remove "sampling and" as written it is redundant.

Line 65 – 67: Your reference to PMF here seems out of place as you do not perform PMF on your results or discuss it as a future method of analysis.

Line 76: Can you really show a sampling campaign? Sampling scheme or setup would be more appropriate.

Line 82: Please clarify your sampling technique here. You say the filter was changed every three hours, but in line 80 you say also performed nine hour sampling during the day.

Line 83: Rephase the nighttime sampling period time from. A clearer phrasing could be "Night-time sampling began at ~17:30 and ended at 08:30 the following day."

Line 88: Was a method blank also performed? If so please explicitly state, if not, please explain how sources of contamination were investigated.

Line 91: What is the purity of these standard compounds? Could there be non-deuterated forms of them within the standard which could contaminate the samples?

Line 102: I'm assuming these were calculated from your deuterated standard compounds. Add a statement saying so. Also, it would be useful to have an idea of the variation of the extraction efficiency as well as the range. Add the mean and standard deviation here if possible. Additionally, if samples were adjusted to account for extraction efficiency that should be stated.

Line 138: Does the simultaneous measurement mean each filter extract was spiked with deuterated compounds, the 16 PAH standard mixture, or the standard curve?

Line 148: This is a very odd way to discuss precision. Why don't you simply state the mean RSD ± σ from all sample replicates of each type? Since your intraday and interday are similar, there may not be any need to state both numbers.

Line 154: Please restructure this. You have not yet introduced systematic error, but this reads as if you've already discussed it.

Line 157:  Systematic error is not uncertainty so you cannot combine it with your precision estimates.  You could propagate the uncertainty associated with the systematic error to better constrain your overall uncertainty.

Line 158:  This, combined with the calculation of extraction efficiencies, is a way to constrain systematic errors.  I recommend combining this section with lines 154-157 and your discussion of extraction efficiencies in 102.

.Line 164:  Add "The number of…has"

Line 166: Rephrase for clarity.  "In China, the official $PM_{2.5}$ annual and 24hr average standards are 35 µg m$^{-3}$ and 75 µg m$^{-3}$, respectively,"

Line 168-169:  Either this statement is incorrect, or your methods section is incorrect.  In your methods section, you state that you've collected 3, 9, and 15 hour averaged samples.  Please fix or clarify.

Line 169:  A more appropriate way of presenting this would be the mean and standard deviation.  "During sampling, the average 24h averaged $PM_{2.5}$ was $X \pm Y$…."

Line 171:  Same as above, the mean and standard deviation would be very meaningful here.

Line 174:  Again, the mean and standard deviation would be very meaningful here.

Line 171 – 178:  All of this information would be presented better in a table.  Consider adding the summary data in this paragraph to the top of each section in table 2.

Line 179 - 193:  Is there any reason why these specific PAHs are important or interesting besides them being found in the highest concentrations?  Some discussion of the particularly interesting compounds would improve the quality of this manuscript.

Line 193:  Can you mention some specific controls the local government has put in place or cite some of the policies?  If they are effective, as you are implying, perhaps they could be applied in other locales, and it would be useful for the reader to know more about them.

Lines 203-213: Is there a specific reason by people report these specific ratios?  Do they mean anything other than the relative amount of each type?  As written, they seem like a random descriptor that isn't adding much meaning to your observations.

207:  Wouldn't the average of the ratio within all the samples be more meaningful.  Or maybe a total ratio of total measured PAH:NPAH?

214: Does daily mean 24 h averaged?  As written the discussion of BaP seems out of place.  Are there specific air quality standards for the other compounds studied?  If so, please discuss them as well.

225:  As written, it's unclear which ratio is being explained after the semicolon.

254: I believe you are trying to say you used an isomeric standard to estimate the sensitivity to a compound where a standard is not available.  Please add some citation explaining why this is

okay with this class of compounds and a statement explaining why you expect the sensitivities to be similar within your analytical setup. Also, if you have measured any other isomer pairs, it might strengthen your case if you show the reader that they have similar sensitivities.

266 – 316:  Correlation can be problematic especially when dealing with interlinked species and non-random sampling. Due to your long averaging times for your filters, and (I'm assuming) relatively quick gas phase measurements it may be difficult to make any concrete conclusions.

278: High CO correlation is not necessarily an indicator of regional sources as you would expect correlations to be high near the sources as well.

272-273:  Please explain what values/units you used in your correlation analysis.  Right now it is unclear if you used the average concentration over the sampling period, the integrated concentration, or something else (i.e., the median).

306-311:  Could this be reversed? Could HONO levels be essential to forming secondary 9-Nitroanthracene?

312: "1/3, 1/9" if you are going to use this symbology to refer to the sampling types, be consistent throughout the paper.

340: Again, some description of the variation of the data is needed.

662:  This data would be better presented in box and whisker plots.

672:  Figure 2 boarders on illegible.  This may have occurred during formatting, but this figure needs to be fixed before publication.

707:  Wouldn't box and whisker plots again be better here as well?

734:  The bars on the plot are very difficult to see.  One way to make this clearer could be to make the width of the bars proportional to the sampling time.

---

## Author Response (AR1)

Response to referee 1:
Author's response in Blue

This is a nicely written paper that contributes new data to help improve understanding of urban sources and levels of PAHs and their derivatives, including day vs night variability.
Thank you for reviewing our paper. Your comments and suggestions were very helpful and we believe the paper has been improved.

What is still missing from the paper in my opinion, is an expression of the study's significance. i.e. what is novel about this study and how/where do the results make an impact on advancing the field of science. I would expect this for a paper in ACP.
To the best of our knowledge, this study is the first to provide data on the organic chemical composition of $PM_{2.5}$ over short time sampling periods (3h) in the city of Beijing. In addition, the statistical relationship between NPAHs and HONO raises the importance of studying the chemical relationship between them, which may impact the HONO budget in the atmosphere and improve related models. Other correlations (CO, NO, $SO_2$) were also important in identifying source emissions.

Author's changes in manuscript, (the following text has been added at the end of the introduction):
This paper explores the feasibility of higher frequency sampling in Beijing, to support the identification of emissions sources from diagnostic ratios and correlations with atmospheric gas pollutants. These measurements also raise the potential importance of the chemical relationship between NPAHs and HONO which may impact the HONO budget in the atmosphere and, if included, improve related models. This study comes after three years of declaring the anti-pollution action plan and strategy taken by the municipal government of Beijing and published in September 2013 (Ministry of Ecology and Environment The People's Republic of China, Beijing toughens pollution rules for cleaner air, 2013), trying to increase the number of days with good air quality index by prohibiting coal combustion, promoting clean energy vehicles and public transport and helping industrial transformation and upgrading to new technologies.

Line 76-86: Why only target PM and not gas-phase? Some explanation is warranted here I think. Also, when comparing results to other studies is this always based on PM and not total?
One of the main objectives of this project (APHH) was to investigate in depth the chemical composition of $PM_{2.5}$ (organics, minerals and ions) and identify emission sources at receptor site. The gas phase measurements of PAHs and derivatives were not a part of this project and the lack of this data can underestimate the exposure risk assessment.
Results were compared to those in the particulate phase in wintertime only. A new study has been added to the comparison (Feng et al., 2019) and text (L.191) modified as follows.

Author's changes in manuscript:
Our average value was comparable to the reported value in a recent study (Feng et al., 2019) at the campus of Peking University health science centre, a short distance from our sampling site (~1 mile), where the authors reported a total PAHs average concentration in winter Beijing (2014 - 2015) of $88.6 \pm 75$ ng m$^{-3}$.

Line 83: Do the longer nighttime samples compared to daytime samples introduce a source of bias in the results? For instance, related to particle capture efficiency which changes as the filter becomes more loaded. Another question is heterogeneous reactions on the filter during the 15hrs the samples are being collected and exposed to a high volume of atmospheric oxidants. If these reactions are occurring they will impact the 15h samples to a greater extent. This issue has been considered in the literature and I think the authors should at least raise this concern.

Data from long sampling times may be biased by sampling artefacts in which an increase in uncertainty on each of the most reactive PAHs compounds may occur due to reactions with gas phase oxidants. Previous studies (Schauer et al., 2003, Goriaux et al., 2006) suggested an underestimation of PAHs concentrations by up to 100% and 200% (under high ozone levels) due to ozone deposition on filters and subsequent chemical decomposition of PAHs. However, at medium ozone levels (30-50 ppb), Schauer et al., 2003, underestimated PAHs values by 30%.

In addition, Tsapakis and Stephanou 2003, have studied under different atmospheric environment (ozone concentration and time sampling) the effect of ozone denuder on gas phase PAHs and particle-bound PAHs. According to this study, ozone affects mostly the gas phase measurement of PAHs (up to 50%) but regarding the particle phase, weaker effect with no apparent difference was suggested at low ozone concentration.

To the best of our knowledge, comparison studies using two different samplers were mostly conducted in Europe where concentrations of PAHs are much lower than in Beijing which may increase the uncertainty. In this study, particles filters were stored at -20°C directly after collection and during transit. The ozone concentrations at night varied between 2 and 21ppb (mean 6.4 ppb, n=16).

Moreover, heterogeneous oxidation processes could occur either in the atmosphere or during the sampling of aerosols and major focus in previous laboratory studies (Ringuet et al., 2012a, Jariyasopit et al., 2014; Zimmermann et al., 2013) was given to the formation of OPAHs and NPAHs. The mechanisms and levels of formation of these derivatives are still far from being fully understood, especially that the reaction of PAHs with oxidants (OH radicals, ozone) is highly influenced by surface coverage and PAHs loading (up to 2 orders of magnitude) (Keyte, I.J., 2013 and references therein) i.e. particles were shown to exhibit a potential inhibiting factor on the reactivity of PAHs due to slow diffusion of oxidants and inaccessibility of PAHs in the bulk particle. Previous studies reported that the formation of NPAHs during high-volume sampling is not significant and calculated to be < 3% (Arey et al., 1988) and < 0.1 % (Dimashki et al., 2000). Therefore, considering the above and our sampling conditions, our estimation of sampling artefacts range between 10 and 20%.

Considering the role of ozone (always below 30 ppb in this study, with a mean value: 10.4 ± 8.8 ppb), in addition to sampling time and temperature, the estimation of the negative sampling artefacts on our data range between 10 and 20 %, with the highest error estimation attributable to longest sampling time (15h).

Line 203-205 and elsewhere: please use consistent number of significant figures when reporting concentrations. In this section it ranges from 2 to 4. I think that either 2 or 3 significant figures is appropriate given method uncertainties.

Concentrations corrected in the manuscript and tables.

Line 369+: I think the spatial variability within Beijing may be even more important than seasonal variability. The authors believe their results are representative of "Beijing" and present them as e.g. "...concentrations for Beijing". However, I wonder if that is appropriate without first investigating the representativeness of their sampling site for the entire metropolis of Beijing. There is also the potential for a dominant impact due to a local source (or sources) that may impact or bias results at a given site. The authors should be careful to address this possibility and not overstate the representativeness of their results (or other results from single sites in other studies), when making comparisons e.g. Beijing vs Athens.

We agree that spatial variability is also important in megacities but most studies focused their discussion on seasonal variability probably because of the lack on data from other districts.

Author's changes in manuscript: (L.193)
The urban location in this study (Fig. S1) was surrounded by busy roads, residential buildings, an underground railway, restaurants and further afield thermal power stations. PAHs concentrations are anticipated to decline closer to the mountains in the North and West of Beijing due to air mass trajectory, aging and distance from emission sources. Results from this study can be considered representative (within the margin of error) of the urban area in Beijing including districts such as Chaoyang, Haidian, Fengtai, Xicheng, Dongcheng, Shijingshan covering an approximate population of 12 million. Future studies in less populated districts and different areas of the metropolitan of Beijing would be helpful for comparison of population exposures.

Response to referee 2:
Author's response in Blue

The authors present a rich data set examining the concentrations of PAH's in atmospheric particulate matter in Beijing. PAHs are known to increase the toxicity of aerosols and this work advances our understanding of their dynamics in the atmosphere. The analytical methodology applied in this work is impressive and carefully reported. Where this manuscript needs improvement is the discussion and presentation of the data. I recommend the authors work to increase the clarity of the results, and their interpretations, to ensure their work is as impactful as possible.
Thank you for your comments, corrections and recommendations. Your careful review and thorough reading helped us to better present the data and we believe the paper has been improved.

**General Comments:**
Comment 1: The authors have a very data rich paper, but often don't report a measure of variability (e.g., standard deviation). This is particularly potent given the title of the manuscript. In the text and in the tables a measure of variability should be included, and discussed, when reporting any data. Specifically, standard deviations should be included in table 2.
The variations of concentrations for each compound were presented in Table 2 as minimum and maximum and standard deviations were presented in Figure 1, which now replaced by box and whiskers plots according to your recommendation below (L.662). The text has been corrected and values of SD for individual compounds and the average total concentration have been added in Table 2, abstract, results and conclusions. In the discussion we have been limited to comparison with Benzo[a]pyrene and total average of PAHs and derivatives. Variations for individual PAHs not frequently reported. We believe the variability of each compound reported in this paper might be helpful for future comparison.

Comment 2: The authors present this work as a higher resolution examination of PAH dynamics because they use relatively high resolution (3h) samples. Despite this, there is almost no discussion of temporal dynamics other than day vs. night or 3h vs. 9h vs. 15 h. I would like to see the authors look into the temporal dynamics on given days and investigate drivers of PAH variability beyond correlation.

This paper explores for the first time the feasibility of higher frequency sampling in Beijing. Whilst we successfully show that this time resolution can be achieved, the relatively short-term data for Beijing has led us to use 24 h averages for comparison and discussion purposes with other work. Temporal dynamics (diurnal, daily, seasonal) of PAHs are influenced by a number of factors such as emission sources, $PM_{2.5}$ concentrations, mixing height level, photolytic effect and gas-particle partitioning. To investigate drivers of PAHs variability beyond correlation and emission sources, we refer primarily to spatial and seasonal variations within the area of sampling. To the best of our knowledge, no high resolution data available in Beijing. The data from summer campaign are under investigation, preliminary results show notably lower average value (~13 times) than the average 24 h total PAHs concentrations in this study (97 ng $m^{-3}$). In summer, concentrations of 2 and 3 ring PAHs were mostly below limit of quantification (in the particle phase) and the average 24 h concentration of B[a]P was $0.8 \pm 2$ ng $m^{-3}$, far lower than the average in winter ($15 \pm 9$ ng $m^{-3}$). PAH partitioning is strongly temperature-dependent; summer time probably also contributes to enhanced photo-degradation of PAHs and photochemical formation of derivatives, and the absence of major sources contributors such as residential heating and combustion. We will add more discussion in our next paper, comparing the temporal dynamics between winter/summer Beijing and Summer New delhi.

It is clear that individual PAHs concentrations are related to $PM_{2.5}$ concentrations as shown in Fig. S2, with contribution from specific emission sources at mid-day. We have reported in this paper that 9-Nitroanthracene appears to accumulate at night and to undergo a photo-degradation during the daytime. We will follow the behaviour of this compound and other component on our ongoing investigation of the summer campaign.

Comment 3: Was there meteorological data available at your study site? PAH lifetime, emission and phase are likely influenced by temperature or solar flux. Additionally, analysis of wind direction trends during the specific emission periods would provide additional insight into the importance of different sources. If this data exists, including it in the analysis could increase the impact of the results

We have reported in the text that no significant correlations were seen between PAHs and meteorological parameters (Relative Humidity and Temperature). Results from correlation of PAHs and derivatives with temperature and relative humidity are shown in Table S1.

The site location in this study was an urban site in Beijing (Fig S1), surrounded by busy traffic roads, residential buildings, underground, restaurants and thermal power stations. PAHs concentrations may fluctuate when getting closer to the mountains in the North and West of Beijing due to air mass trajectory, aging or moving away from emission sources. A wind-rose plot of the entire campaign is presented in Fig S4, showing that winds arriving at the site blow from the North East for much of the time. Contribution to gas phase concentrations of $SO_2$ (L281-284) was attributed to a specific anthropogenic source such as the Beijing Taiyanggong thermal power station in the North East.

Comment 4: Organization. This manuscript, although data rich, is poorly organized with many acronyms, lots of in text data, and unclear conclusions. I recommend the authors make an attempt to trim the text. Specifically, many data references in the discussion could be replaced with reference to the appropriate figure or table. This will allow the readers to better understand the author's conclusions by cleaning up the text.

Acronyms were used only to shorten target compounds full name in the text and they are listed in Table 1. Acronyms in the conclusion has been replaced by the full name of each compound used to clarify the information to the reader without referring to Table 1.

References to appropriate Tables and Figures have been added to the discussion

Author's changes in manuscript:
In the conclusions:
(L.388) Benzo[a]pyrene; (L.393) 2+3Nitrofluoranthene/1-Nitropyrene; (L.400) 9-Nitroanthracene.
In the discussion:
- (L.232) As shown in Fig. 4, other ratios can be useful
- (L.271) parameters (Relative Humidity and Temperature) as shown in Table S1
- (L.292) For NPAHs, as shown in Table S1, no significant correlation
- (L.307) As shown in Table S2, 9-Nitroanthrancene
- (L.340) BaP$_{eq}$ for the whole sampling period was 23.6 ng m$^{-3}$ (Table 3).
- (L.340) As shown in Table 2, 6-NCHR has not been quantified
- (L.344) in urban air of Beijing ranged from 10$^{-5}$ to 10$^{-3}$ > 10$^{-6}$ (Table 3).
- (L.349) the 24 h average estimated cancer risk (Table 3) from inhalation exposure to

Comment 5: The authors could improve their argument for the specific significance of their work. The authors do a good job of detailing PAHs in aerosol but don't discuss their site in significant detail. Please elaborate on why there is a pressing need to study PAHs in Beijing within your introduction

New text addressing this point has been added at the end of the introduction section.

Author's changes in manuscript:
This paper explores the feasibility of higher frequency sampling in Beijing, to support the identification of emissions sources from diagnostic ratios and correlations with atmospheric gas pollutants. These measurements also raise the potential importance of the chemical relationship between NPAHs and HONO which may impact the HONO budget in the atmosphere and, if included, improve related models. This study comes after three years of declaring the anti-pollution action plan and strategy taken by the municipal government of Beijing and published in September 2013 (Ministry of Ecology and Environment The People's Republic of China, Beijing toughens pollution rules for cleaner air, 2013), trying to increase the number of days with good air quality index by prohibiting coal combustion, promoting clean energy vehicles and public transport and helping industrial transformation and upgrading to new technologies.

Comment 6: For a field study, site selection and significance is very important. Please add some discussion of how representative your site is compared to Beijing in general, what the strengths of the site are, and what are some possible weaknesses.

Most studies focused their discussion on seasonal variability probably because of the lack on data from other districts. New text has been added to the results and discussion

Author's changes in manuscript (Section 3.1 - L. 193):
The urban location in this study (Fig. S1) was surrounded by busy roads, residential buildings, an underground railway, restaurants and further afield thermal power stations. PAHs concentrations are anticipated to decline closer to the mountains in the North and West of Beijing due to air mass trajectory, aging and distance from emission sources. Results from this study can be considered representative (within the margin of error) of the urban area in Beijing including districts such as Chaoyang, Haidian, Fengtai, Xicheng, Dongcheng, Shijingshan covering an approximate population of 12 million. Future studies in less populated districts and different areas of the metropolitan of Beijing would be helpful for comparison of population exposures.

**Specific Comments**

Line 20: Restructure the list of compounds. As written, it is easy to miss the period after the major PAHs and not realize a new list has started for the major OPAHs. Use the same sentence structure for each type to avoid confusing the reader.

Corrected

Line 38: comma after "organic carbon"

Corrected

Line 58: Clarify "geographic peripheral expansion"

Corrected

Author's changes in manuscript (underlined text):

a major focus has been given to Chinese cities such as Shanghai, Beijing, Guangzhou, Tianjin, and Shenzhen because of their population growth and geographic peripheral expansion in manufacturing capacity and energy industries which are located throughout each of the city's manufacturing zones.

Line 60: change "PAH on 24 PM2.5 sampling" to "PAH concentrations in 24 h averaged samples".

Corrected

Author's changes in manuscript:

The majority of previous studies have reported PAH concentrations in 24 h averaged samples during short-term and long-term measurements campaigns

Line 62: Remove "sampling and" as written it is redundant.

Removed

Line 65 – 67: Your reference to PMF here seems out of place as you do not perform PMF on your results or discuss it as a future method of analysis.

We did mention the PMF model as an example because it is widely used in parallel with diagnostic ratios.

Line 76: Can you really show a sampling campaign? Sampling scheme or setup would be more appropriate.

Corrected

Line 82: Please clarify your sampling technique here. You say the filter was changed every three hours, but in line 80 you say also performed nine hour sampling during the day.
Author's changes in manuscript: (L.82)
The daytime sampling started at 8:30 in the morning and the filter was changed every 3 h. During low particulate loading conditions, the daytime sampling started at 8:30 in the morning for a sampling duration of 9 h.

Line 83: Rephrase the nighttime sampling period time from. A clearer phrasing could be "Night-time sampling began at ~17:30 and ended at 08:30 the following day."
Author's changes in manuscript:
Night-time sampling began at ~ 17:30 and ended at 08:30 the following day.

Line 88: Was a method blank also performed? If so please explicitly state, if not, please explain how sources of contamination were investigated.
New text added to Data analysis and error evaluation (L.145)
Author's changes in manuscript:
To determine any sources of contamination during sample preparation and the analytical procedure, the solvent (acetonitrile) and field blanks (n=2) were analysed following the same procedure as for the samples (Extraction, SPE, Evaporation). Most target compounds were found to be below LOD (S/N=3) or orders of magnitude (up to $10^3$- $10^4$) lower than was found in the samples. A small number of compounds found in field blanks (1,8-Naphthalic anhydride, Benzo[a]fluorenone, 1-Nitronaphthalene, 9-Nitroanthracene) have a higher contribution (4-30 %) to very few filters (2 to 5 samples) collected over a 3 h time period, if this was co-incident with low particulate loading conditions. The contribution to each compound from field blanks has been corrected in the final data.

Line 91: What is the purity of these standard compounds? Could there be non-deuterated forms of them within the standard which could contaminate the samples?
Minimum purity of D2-standard compounds was 98%. Results from clean filters spiked with deuterated-compounds did not show contribution on our target compounds.

Line 102: I'm assuming these were calculated from your deuterated standard compounds. Add a statement saying so. Also, it would be useful to have an idea of the variation of the extraction efficiency as well as the range. Add the mean and standard deviation here if possible. Additionally, if samples were adjusted to account for extraction efficiency that should be stated.
We have adjusted concentrations to recovery efficiencies and it was stated in the text (L.103)
Author's changes in manuscript:
The average recovery efficiencies calculated from surrogate standards ranged from 85% to 96% (Phenanthrene-d10: 95 ± 9 %; Pyrene-d10: 101 ± 7 %; 9-Fluorenone-d8: 98 ± 13 %; 9,10-Anthraquinone-d8: 102 ± 11 %; 1-Nitronaphthalene-d7: 93 ± 8 %; 3-Nitrofluoranthene-d9: 101 ± 11 %) and the target compounds concentrations were calculated incorporating measured recovery efficiencies.

Line 138: Does the simultaneous measurement mean each filter extract was spiked with deuterated compounds, the 16 PAH standard mixture, or the standard curve?

Filters were spiked before extraction. The calibration standards (16 PAHs standard mix and deuterated mix) were measured every day during the sequence analysis of the samples in order to follow the response of the instrument and the drift of the calibration curve.

Line 148: This is a very odd way to discuss precision. Why don't you simply state the mean RSD $\pm \sigma$ from all sample replicates of each type? Since your intraday and interday are similar, there may not be any need to state both numbers.

As we didn't use the standard reference materials from EPA as a reference for method validation, we think it is important to show the response of the instrument for each compound in replicates samples between days in order to include the average RSD in the error estimation.

Line 154: Please restructure this. You have not yet introduced systematic error, but this reads as if you've already discussed it.

There is no particular introduction for systematic error as they are different for each type of study and measurement. In this study, we have related the systematic error to the influence of the sample matrix during the analysis sequence on the quantification step and the calibration offset (L.154-155).

Line 157: Systematic error is not uncertainty so you cannot combine it with your precision estimates. You could propagate the uncertainty associated with the systematic error to better constrain your overall uncertainty.

Corrected, "uncertainty" removed

Author's changes in manuscript (L.157):
Therefore, the overall estimated error, combining the precision and the systematic effects, is less than 20%.

Line 158: This, combined with the calculation of extraction efficiencies, is a way to constrain systematic errors. I recommend combining this section with lines 154-157 and your discussion of extraction efficiencies in 102.

This section has been modified and combined with line 145 please refer to the response to question (L.88) above.

Line 164: Add "The number of…has"
Corrected

Line 166: Rephrase for clarity. "In China, the official PM2.5 annual and 24hr average standards are 35 µg m-3 and 75 µg m-3, respectively,"
Author's changes in manuscript:
In China, the official air quality guidelines for $PM_{2.5}$ expressed as annual mean and 24 h average limit are 35 µg m$^{-3}$ and 75 µg m$^{-3}$, respectively.

Line 168-169: Either this statement is incorrect, or your methods section is incorrect. In your methods section, you state that you've collected 3, 9, and 15 hour averaged samples. Please fix or clarify.

$PM_{2.5}$ concentrations were averaged to 24 h for comparison with the official guidelines, also they were averaged to the filter sampling times and presented in Fig. 6 and Fig. S3

Author's changes in manuscript (L.170):
Concurrent $PM_{2.5}$ concentrations were averaged to the filter sampling times (3 h, 9 h, 15 h) and are shown in Fig.6 and Fig. S3.

Line 169: A more appropriate way of presenting this would be the mean and standard deviation. "During sampling, the average 24h averaged PM2.5 was X ± Y...."
Author's changes in manuscript (L.170):
The average 24 h concentration was $108 \pm 82$ µg m$^{-3}$ (range: 10 - 283 µg m$^{-3}$), exceeding the 24 h limit value on 10 of the 18 sampling days.

Line 171: Same as above, the mean and standard deviation would be very meaningful here.
Corrected

Line 174: Again, the mean and standard deviation would be very meaningful here.
Corrected

Line 171 – 178: All of this information would be presented better in a table. Consider adding the summary data in this paragraph to the top of each section in table 2.
Concentrations of each compound and the summary data (total average) are presented in Table 2, and new Fig. 1.

Line 179 - 193: Is there any reason why these specific PAHs are important or interesting besides them being found in the highest concentrations? Some discussion of the particularly interesting compounds would improve the quality of this manuscript.
These specific 16 PAHs are considered representatives for all PAHs and they are defined by the United States Environment Protection Agency (EPA) (L.105). They have known toxicity equivalence factors (Table S3) and considered carcinogenic to human health. The particularity of these compounds and their presence in the particulate phase were discussed in paragraph diagnostic ratios to identify emission sources (L.218). Other particularities were suggested to anthracene and 9-Nitroanthracene (L.306-316).

Line 193: Can you mention some specific controls the local government has put in place or cite some of the policies? If they are effective, as you are implying, perhaps they could be applied in other locales, and it would be useful for the reader to know more about them.
Author's changes in manuscript:
Introduction (L.73): This study comes after three years of declaring the anti-pollution action plan and strategy taken by the municipal government and published in September 2013

(Ministry of Ecology and Environment The People's Republic of China, Beijing toughens pollution rules for cleaner air, 2013), trying to increase the number of days with good air quality index by prohibiting coal combustion, promoting clean energy vehicles and public transport and helping industrial transformation and upgrading to new technologies.

Results and discussion section 3.1 (L.194):
Our average value (97 ng m$^{-3}$) was comparable to the reported value in a recent study (Feng et al., 2019) at the campus of Peking University health science centre, a short distance from our sampling site (~1 mile), where the authors reported a total PAHs average concentration in winter Beijing (2014 - 2015) of 88.6 ± 75 ng m$^{-3}$. The lower average concentration of total PAHs reported in this study and Feng et al., (2019) can potentially be attributed to the efforts from municipal government to improve air quality and control emissions by reducing combustion sources.

Lines 203-213: Is there a specific reason by people report these specific ratios? Do they mean anything other than the relative amount of each type? As written, they seem like a random descriptor that isn't adding much meaning to your observations.
The use of these ratios shows the possible contribution of PAHs to OPAH and NPAH, the observed levels simplify the comparison with other studies. In other words, even if the concentration of total PAH is much higher in China than Europe, using these specific ratios facilitate the comparison, and highlight the chemical transformation under similar conditions such as winter.

207: Wouldn't the average of the ratio within all the samples be more meaningful. Or maybe a total ratio of total measured PAH:NPAH?
Previous studies have reported the total 24 h average for each series of compound. The use of the average 24 h is more meaningful here and represents the mean concentration of all samples and these ratios represent the total average concentration of each series of compound. The symbol "∑" added to the text.

214: Does daily mean 24 h averaged? As written the discussion of BaP seems out of place. Are there specific air quality standards for the other compounds studied? If so, please discuss them as well.
It means 24 h and it was compared with the 24 h average limit value for China. This paragraph was moved up to the beginning of this section (L.170). "24 h" added to the text.
According to WHO, BaP is a suitable marker of particle-bound PAH and typically represents a substantial proportion of the total carcinogenic potential of the parent PAH.

Author's changes in manuscript:
The daily (24 h) concentration of Benzo[a]pyrene ranged from 4.46 to 29.8 ng m$^{-3}$ (average 15 ± 8.9 ng m$^{-3}$)

225: As written, it's unclear which ratio is being explained after the semicolon.
Author's changes in manuscript: (L.225)
values of FLT/(FLT + PYR) less than 0.4 and IcdP/(IcdP + BghiP) less than 0.2, are mostly related to incomplete combustion.

254: I believe you are trying to say you used an isomeric standard to estimate the sensitivity to a compound where a standard is not available. Please add some citation explaining why this is okay with this class of compounds and a statement explaining why you expect the sensitivities to be similar within your analytical setup. Also, if you have measured any other isomer pairs, it might strengthen your case if you show the reader that they have similar sensitivities.

This approach was based on previous studies cited in L.256-258. Any recommended article from the referee would be very helpful. It is not necessarily that isomers show the same sensitivities and the best way to check that is to analyse the isomer pairs. Using a Q-TOF-MS as in our study helped bring the sensitivity of isomer to the same level by taking into account transition and qualifier ions. PAH isomers used in this study showed relatively same sensitivities.

Author's changes in manuscript: (L.255)
PAH isomer pairs (Table 1) in standard mixtures showed similar sensitivities for each concentration used, therefore, we assume an equal sensitivity for 2-NFLT and 3-NFLT during analysis.

– 316: Correlation can be problematic especially when dealing with interlinked species and non-random sampling. Due to your long averaging times for your filters, and (I'm assuming) relatively quick gas phase measurements it may be difficult to make any concrete conclusions.

It is true that the use of correlation between the gas phase measurements of atmospheric oxidants and the chemical composition of the particle phase does not prove a chemical link but it can support a chemical reaction pathway such as the positive strong correlation between ANT and 9-NANT in particle phase. This suggestion supports the heterogeneous formation pathway of 9-NANT (in parallel, ANT gas phase to 9-NANT particle), which mean 9-NANT is closely related to ANT and not to direct emissions.

278: High CO correlation is not necessarily an indicator of regional sources as you would expect correlations to be high near the sources as well.
Yes, but we cannot exclude regional emissions due to the long lifetime of CO.

272-273: Please explain what values/units you used in your correlation analysis. Right now it is unclear if you used the average concentration over the sampling period, the integrated concentration, or something else (i.e., the median).
Fig. 5 shows an example of the unit values and type of data used for correlation. The concentrations (ng m$^{-3}$) used are the total for each class of compounds (PAHs, OPAHs, NPAHs) in each sample (3h, 9h, 15h). Concentrations of the gas phase species were in ppt/ppb levels and have been time-averaged to the filter sampling times. There is no need to show units to test the correlation of significance between two variables.

306-311: Could this be reversed? Could HONO levels be essential to forming secondary 9-Nitroanthracene?

The loss rate of HONO on dust aerosols was shown to be negligible ($k' = 4 \times 10^{-9}$, El zein et al., 2013) and not important in comparison with its photolysis rate in daytime $\sim 1.3 \times 10^{-3}$ s$^{-1}$ or deposition at night-time. Therefore, the adsorption of HONO on aerosols and its contribution to products formation either during the day or the night-time can be considered as negligible.
Reference: El Zein, A.; Romanias, M.; Bedjanian, Y.: Kinetics and Products of Heterogeneous Reaction of HONO with $Fe_2O_3$ and Arizona Test Dust. Environ. Sci. Technol. 2013, 47, 6325−6331

312: "1/3, 1/9" if you are going to use this symbology to refer to the sampling types, be consistent throughout the paper.
Removed from the manuscript.

340: Again, some description of the variation of the data is needed.
Corrected

662: This data would be better presented in box and whisker plots.
Figure 1 modified to box and whisker plots

672: Figure 2 boarders on illegible. This may have occurred during formatting, but this figure needs to be fixed before publication.
Figure 2 modified and re-arranged

707: Wouldn't box and whisker plots again be better here as well?
We think this figure is the best way to show this information to the reader, with no need for box plots

734: The bars on the plot are very difficult to see. One way to make this clearer could be to make the width of the bars proportional to the sampling time.
Figure 6 modified

[revised manuscript text omitted]

**Commented [A30]:** The symbol ''∑'' added to the text in response to Referee 2: Specific comments: L.207

**Commented [A31]:** Response to Referee 2: Specific comments: L.225

**Commented [A32]:** Response to Referee 2: Comment 4

**Commented [A33]:** Response to Short Comment (SC1)

[revised manuscript text omitted]

**Commented [A54]:** Response to Referee 2: Specific comments: L.734